# Understanding the origin of Paris Agreement emission uncertainties

Joeri Rogelj[1,2], Oliver Fricko[1], Malte Meinshausen[3,4], Volker Krey[1], Johanna J.J. Zilliacus[1] & Keywan Riahi[1,5]

The UN Paris Agreement puts in place a legally binding mechanism to increase mitigation action over time. Countries put forward pledges called nationally determined contributions (NDC) whose impact is assessed in global stocktaking exercises. Subsequently, actions can then be strengthened in light of the Paris climate objective: limiting global mean temperature increase to well below 2 °C and pursuing efforts to limit it further to 1.5 °C. However, pledged actions are currently described ambiguously and this complicates the global stocktaking exercise. Here, we systematically explore possible interpretations of NDC assumptions, and show that this results in estimated emissions for 2030 ranging from 47 to 63 $GtCO_2e\,yr^{-1}$. We show that this uncertainty has critical implications for the feasibility and cost to limit warming well below 2 °C and further to 1.5 °C. Countries are currently working towards clarifying the modalities of future NDCs. We identify salient avenues to reduce the overall uncertainty by about 10 percentage points through simple, technical clarifications regarding energy accounting rules. Remaining uncertainties depend to a large extent on politically valid choices about how NDCs are expressed, and therefore raise the importance of a thorough and robust process that keeps track of where emissions are heading over time.

[1] ENE Program, International Institute for Applied Systems Analysis (IIASA), Schlossplatz 1, Laxenburg A-2361, Austria. [2] Institute for Atmospheric and Climate Science, ETH Zurich, Universitätstrasse 16, Zurich 8006, Switzerland. [3] Australian-German College of Climate & Energy Transitions, The University of Melbourne, Parkville, Victoria 3010, Australia. [4] Potsdam Institute for Climate Impact Research (PIK), Telegraphenberg A26, Potsdam 14412, Germany. [5] Graz University of Technology, Inffeldgasse, Graz A-8010, Austria. Correspondence and requests for materials should be addressed to J.R. (email: rogelj@iiasa.ac.at).

The United Nations Framework Convention on Climate Change's (UNFCCC) Paris Agreement[1] has been heralded as a critical step in the global response to the threat of climate change. It establishes a long-term temperature goal of holding global mean temperature increase well below 2 °C and pursuing efforts to limit it to 1.5 °C relative to preindustrial levels. This climate goal is accompanied by a legally binding review architecture in which countries submit national climate plans every five years (although the pledges themselves are not legally binding)[2,3]. These so-called nationally determined contributions (NDCs) cover aspects of mitigation and adaptation, together with issues related to means of implementation (for example, capacity building, international finance and technology transfer), comparability and fairness, or sometimes linkages to sustainable development[4]. Alternating with the five-yearly submission cycle of NDCs, periodical stocktaking exercises of implementation progress will be carried out by the Parties to the Paris Agreement. These stocktaking exercises will assess the collective progress towards the achievement of the agreement's goals, and the Paris Agreement explicitly highlights that they will be carried out in light of the best available science.

The NDCs constitute a core element of the Paris Agreement's mitigation architecture. An essential characteristic of these NDCs is that they are determined at the national level, and for the first round of submissions this was taken very literally. Virtually every aspect of the submitted NDCs was decided nationally, and little to no guidance or requirements were given that could focus their scope or enable comparability and quantifications of pledged actions. This led published estimates of the overall emission implications of current NDCs until 2030 to vary widely[3–8]. For example, a recent re-analysis of ten literature studies found a range of 49–58 billion tonnes of annual $CO_2$ equivalent greenhouse gas (GHG) emissions ($GtCO_2e\ yr^{-1}$, in ref. 3 GHG emissions were aggregated with 100-year Global Warming Potentials, GWP-100, from ref. 9) for projected emissions in 2030 under various interpretations of the NDCs. Several drivers of this uncertainty have been suggested, but a systematic exploration of potential drivers and a quantification of their influence is lacking.

Here we focus on understanding the mitigation aspects of NDCs and carry out a systematic analysis of one of the key components of the abovementioned stocktaking exercises: estimating implied GHG emissions under the current intended NDCs and assessing potential sources of uncertainty. We also provide a first-order assessment of longer term implications of the NDCs for the Paris Agreement temperature goal. We find that global emissions in line with current NDCs can vary by − 10% to + 20% around our global median estimate of 52 $GtCO_2e\ yr^{-1}$ in 2030. A further decomposition of this uncertainty shows that socioeconomic baseline uncertainties dominate, followed by variations in alternative energy accounting methods. The overall range can be reduced by technical clarifications, but part of this uncertainty results from political choices about how NDCs are formulated. For example, some NDCs are expressed as emissions intensity improvements. The latter source of uncertainty does not come with a simple technical fix. Finally, we show how post-2030 challenges to limit global mean temperature rise to below 1.5 °C or 2 °C vary strongly as a function of the emissions uncertainty range in 2030.

## Results

**Exploration of six uncertainty dimensions**. We quantify how varying the interpretation of NDCs in six uncertainty dimensions can influence the spread in GHG emissions estimates. An overview of the six uncertainty dimensions covered here is provided in Table 1 and Fig. 1. Two dimensions are linked to uncertainties in socioeconomic baseline developments until 2030 and variations in historical emissions estimates. Two further dimensions are directly linked to the formulation of NDC targets, like the conditionality of NDCs targets, or the expression of NDC targets as a range instead of a single number. The final two dimensions are linked to how renewable energy is accounted for in NDCs, and whether non-commercial biomass energy is counted towards renewable energy. Importantly, we explore emissions uncertainty implied by the current NDCs but do not attempt to predict where global GHG emissions are heading based on current national policies and measures.

We analyse the influence of the six abovementioned dimensions (Table 1) on both regional and global emissions estimates for 2030 in a global, integrated framework with a detailed representation of the energy-economic system. Any NDC modelling exercise has to make tradeoffs between the detail of the representation of national policies and the inclusion of global feedbacks and feedbacks between regions. We use a global model that allows for macroeconomic interactions and feedbacks between the NDCs in various regions. This is crucial, as it allows for a globally consistent analysis unlike stacked bottom-up assessments of national models. We created a set of 144 scenarios, each with a different interpretation of a certain aspect of the NDCs (Fig. 1, see Methods for further details).

**Key determinants of global NDC uncertainty**. Implementing the current NDCs (Supplementary Data 1, Methods) under varying assumptions for our six uncertainty dimensions (Table 1 and Fig. 1) results in a range of 47–63 $GtCO_2e\ yr^{-1}$ of potential GHG emissions in 2030 (Fig. 2, Table 2, in this study $CO_2$-equivalence is always aggregated with GWP-100 values from the IPCC AR4 (ref. 10) unless stated otherwise). Even when correcting for the difference in $CO_2$-equivalence metric, this range spans the multi-model range of studies available in the literature[3] (Supplementary Table 1).

Global emissions in the year 2010 are ∼51 $GtCO_2e\ yr^{-1}$ in our model, and year 2014 emissions are independently estimated[11] at 54 $GtCO_2e\ yr^{-1}$ (90% uncertainty range of about ±10%)[12]. This implies that the current ambiguity in NDC formulations can result in emissions that either continue to increase, stabilize, or decrease by 2030, depending on their interpretation. Because the Paris Agreement also aims at reaching global peaking of GHG emissions as soon as possible[1], the question whether year-2030 emissions are higher or lower than today under the NDCs is a critical one for the global stocktake.

Our analysis allows us to further explore the emissions uncertainty range. The dominant driver of the uncertainty at the global scale is the potential variation in socioeconomic assumptions (Fig. 3b). Estimates of global GHG emissions under the NDCs in 2030 vary by ∼15–20% relative to the median of our estimates, depending on the overall socioeconomic development over the next 15 years. We model variations in socioeconomic baselines by assuming implementations of different Shared Socioeconomic Pathways[13] (SSPs), representing a quantification of a sustainable green-growth world (SSP1), a world with regional rivalry (SSP3), and a middle-of-the-road world following historical experience (SSP2; see also Fig. 1 and Table 1). For example, across these three SSPs global economic growth between 2020 and 2030 varies from 3.6 to 5.3% $yr^{-1}$, while total final energy intensity improvements range from 0.3 to 1.7% $yr^{-1}$ depending on the SSP storylines (see ref. 14 and ref. 15 for detailed information about GDP projections). Other uncertainty sources, like the conditionality of NDC targets, alternative energy equivalence accounting methods for renewable

**Table 1 | Overview of explanation and implementation of assessed NDC uncertainty dimensions.**

| Explanation | Implementation | Affected regions* (top 3) |
|---|---|---|
| *Socioeconomic baseline variation*<br>Variations in assumed future socioeconomic development which influence NDC emission estimates. For example, when actions are specified as carbon intensity improvements (the reduction of $CO_2$ emissions per unit of economic output), or relative to an unspecified hypothetical baseline in absence of climate mitigation policies and measures. | NDCs are assessed under three socioeconomic futures from the SSPs. These three futures represent a quantification of a sustainable green-growth world (SSP1), a world with regional rivalry (SSP3), and a middle-of-the-road world following historical experience (SSP2), as described in ref. 14. | Centrally Planned Asia (CPA)<br>South Asia (SAS)<br>Former Soviet Union (FSU) |
| *Historical emission variation*<br>Variations in historical emission inventories influence NDC emission estimates when NDC objectives are specified as a percentage change from a historical value, or when no-policy baselines are started from a historical value. | NDCs are assessed under three different historical emission data sets[11,42,43]. | Sub-Saharan Africa (AFR)<br>Latin-America (LAM)<br>Former Soviet Union (FSU) |
| *Conditionality*<br>Some NDC actions come with conditions attached to them, for example regarding the availability of finance[3]. Whether these conditions will be met is uncertain. | Two cases, one in- and the other excluding conditional actions, are assessed. | Pacific Asia (PAS)<br>Sub-Saharan Africa (AFR)<br>Latin-America (LAM) |
| *Range specifications*<br>Instead of providing one single target number, some NDCs propose a target range. | Two cases, one with the minimum and the other with the maximum of each respective NDC target range, are assessed. | Centrally Planned Asia (CPA)<br>Sub-Saharan Africa (AFR) (North America and Pacific OECD, NAM and PAO, but small absolute variations) |
| *Alternative energy accounting methods*<br>The contributions of renewable and fossil energy sources can be compared by expressing renewable energies in 'primary energy equivalence'. Several methods exist to make this conversion. This influences emission estimates if NDCs target to achieve a specific share of renewable energies in the energy mix. | NDCs are assessed assuming two primary energy equivalence methods: the direct equivalence method, and the partial substitution method. | Centrally Planned Asia (CPA) |
| *Attribution of non-commercial biomass*<br>Non-commercial biomass covers an important share of the overall energy demand in some regions.[50] Whether this non-commercial biomass is counted towards renewable energies can influence how easily a country can meet an NDC target which aims at achieving a specific share of renewable energies in the energy mix. | Two cases, one where non-commercial biomass is counted towards renewable primary energy and one where it is not, are assessed. | Centrally Planned Asia (CPA) |

GHG, greenhouse gas; NDC, nationally determined contribution; SSP, shared socioeconomic pathways.
*Regions are defined in Supplementary Table 2.

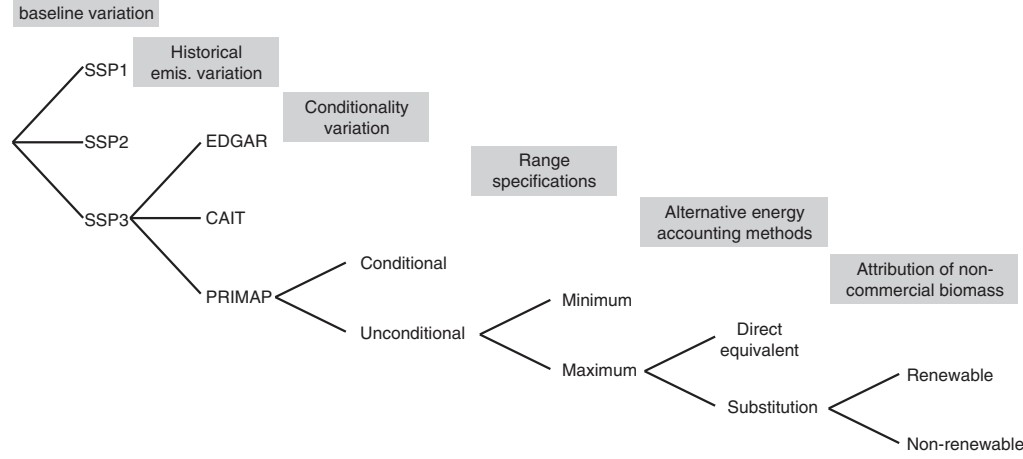

**Figure 1 | Overview of scenario structure to explore six uncertainty dimensions.** A total of 3*3*2*2*2 = 144 scenarios has been developed.

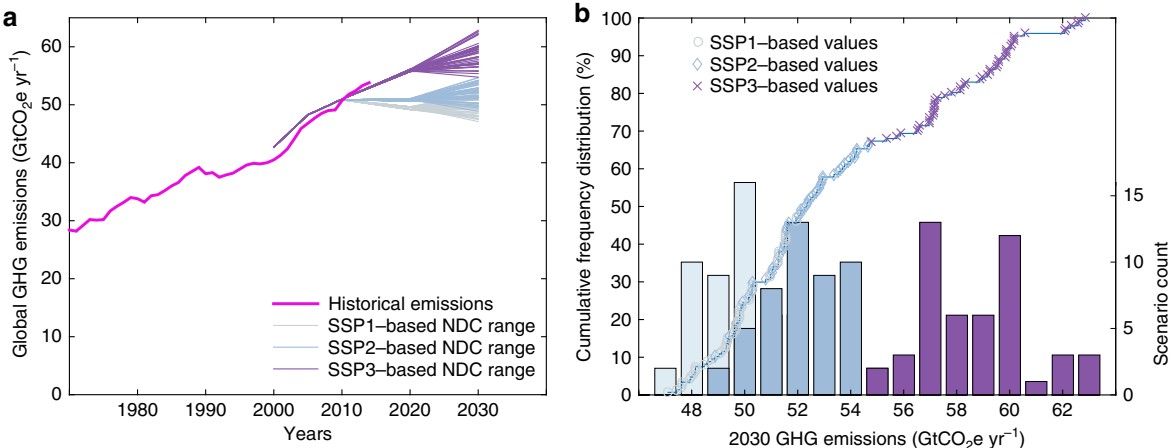

**Figure 2 | Range of 2030 emissions resulting from various interpretations of the current NDCs.** (**a**) Global historical GHG emissions from ref. 11 and projected emissions under the current NDCs. Each line from 2010 to 2030 represents one of 144 modelled scenarios; (**b**) Cumulative distribution (line with individual scenario symbols on top, left axis) and histograms (bars, right axis) of GHG emission estimates for 2030 under the NDCs. Projections are grouped based on the SSP on which their baseline assumptions are based.

**Table 2 | Estimated impact of assessed uncertainty dimensions on 2030 GHG emissions.**

| Global* | AFR | CPA | EEU | FSU | LAM | MEA | NAM | PAO | PAS | SAS | WEU |
|---|---|---|---|---|---|---|---|---|---|---|---|
| *Mean emission estimate (GtCO$_2$e yr$^{-1}$) in 2030* | | | | | | | | | | | |
| 53.5 | 3.4 | 14.4 | 1.0 | 3.8 | 5.2 | 4.1 | 6.1 | 1.9 | 4.2 | 5.8 | 3.8 |
| *Median emission estimate (GtCO$_2$e yr$^{-1}$) in 2030* | | | | | | | | | | | |
| 52.3 | 3.4 | 13.3 | 1.0 | 3.9 | 5.2 | 4.0 | 6.1 | 1.9 | 4.1 | 5.4 | 3.8 |
| *Overall emission estimate incl. uncertainty†* | | | | | | | | | | | |
| 47.1–62.9 | 2.8–4.2 | 11.0–20.5 | 0.9–1.0 | 3.3–4.5 | 4.7–5.7 | 3.6–4.7 | 5.9–6.2 | 1.9–1.9 | 3.9–4.5 | 5.1–6.9 | 3.8–3.8 |
| *Uncertainty due to socioeconomic baseline variation†* | | | | | | | | | | | |
| 7.1–11.3 | 0.1–0.4 | 3.4–7.2 | 0–0 | 0.4–1.2 | 0.1–0.7 | 0.6–0.7 | 0–0 | 0–0 | 0.1–0.2 | 1.7–1.8 | 0–0 |
| *Uncertainty due to historical emission variation†* | | | | | | | | | | | |
| 0.1–1.2 | 0.1–0.3 | 0–1.3 | 0–0 | 0–0.6 | 0.1–0.5 | 0–0.2 | 0.1–0.2 | 0–0.1 | 0–0.1 | 0–0.1 | 0–0 |
| *Uncertainty due to conditionality of NDCs†* | | | | | | | | | | | |
| 1.0–2.7 | 0.4–0.8 | 0–0.4 | 0–0 | 0–0.1 | 0–0.8 | 0.2–0.5 | 0–0 | 0–0 | 0.4–0.5 | 0–0.1 | 0–0 |
| *Uncertainty due to range specifications of NDCs†* | | | | | | | | | | | |
| 0.3–3.1 | 0.1–0.4 | 0–2.4 | 0–0 | 0–0.2 | 0–0.1 | 0–0 | 0.1–0.1 | 0–0 | 0–0 | 0–0 | 0–0 |
| *Uncertainty due to alternative energy accounting methods†* | | | | | | | | | | | |
| 0–4.5 | 0–0 | 0–4.5 | 0–0 | 0–0.1 | 0–0.1 | 0–0 | 0–0 | 0–0 | 0–0 | 0–0.1 | 0–0 |
| *Uncertainty due to attribution of non-commercial biomass†* | | | | | | | | | | | |
| 0–1.7 | 0–0 | 0–1.8 | 0–0 | 0–0 | 0–0 | 0–0 | 0–0 | 0–0 | 0–0 | 0–0 | 0–0 |

GHG, greenhouse gas; NDC, nationally determined contribution.
Supplementary Table 1 provides values aggregated with values from ref. 9.
*Regions are defined in Supplementary Table 2 and illustrated in Fig. 3.
†Uncertainty ranges are minimum–maximum ranges (Methods) in GtCO$_2$e yr$^{-1}$ (aggregated with GWP-100 values from ref. 10).

energy targets, or the fact that some NDCs provide a range instead of a single specific target all imply variations of less than 10% around our median estimate (Fig. 3b, Table 2, uncertainties are always reported relative to the median estimate, unless stated otherwise).

**Regional variations.** Not all regions contribute equally to the overall uncertainty (Fig. 3a,c) and dominant drivers differ across regions (Fig. 3d). First of all, regions of which ambiguous NDC specifications result in the highest degree of uncertainty in

regional emissions estimates are Centrally Planned Asia (CPA, dominated by China), South Asia (SAS, dominated by India), Sub-Saharan Africa (AFR) and the Former Soviet Union (FSU, dominated by the Russian Federation), followed by the Middle East and Northern Africa (MEA), Latin America (LAM) and Pacific Asia (PAS) (Fig. 3a,d and Supplementary Table 2 for an overview of regional definitions). Regions that most often define NDC emission reductions in terms of a percentage reduction below a historical base year, like Eastern and Western Europe (EEU and WEU, respectively), North America (NAM, dominated by the USA) or Pacific OECD countries (Japan, Australia and

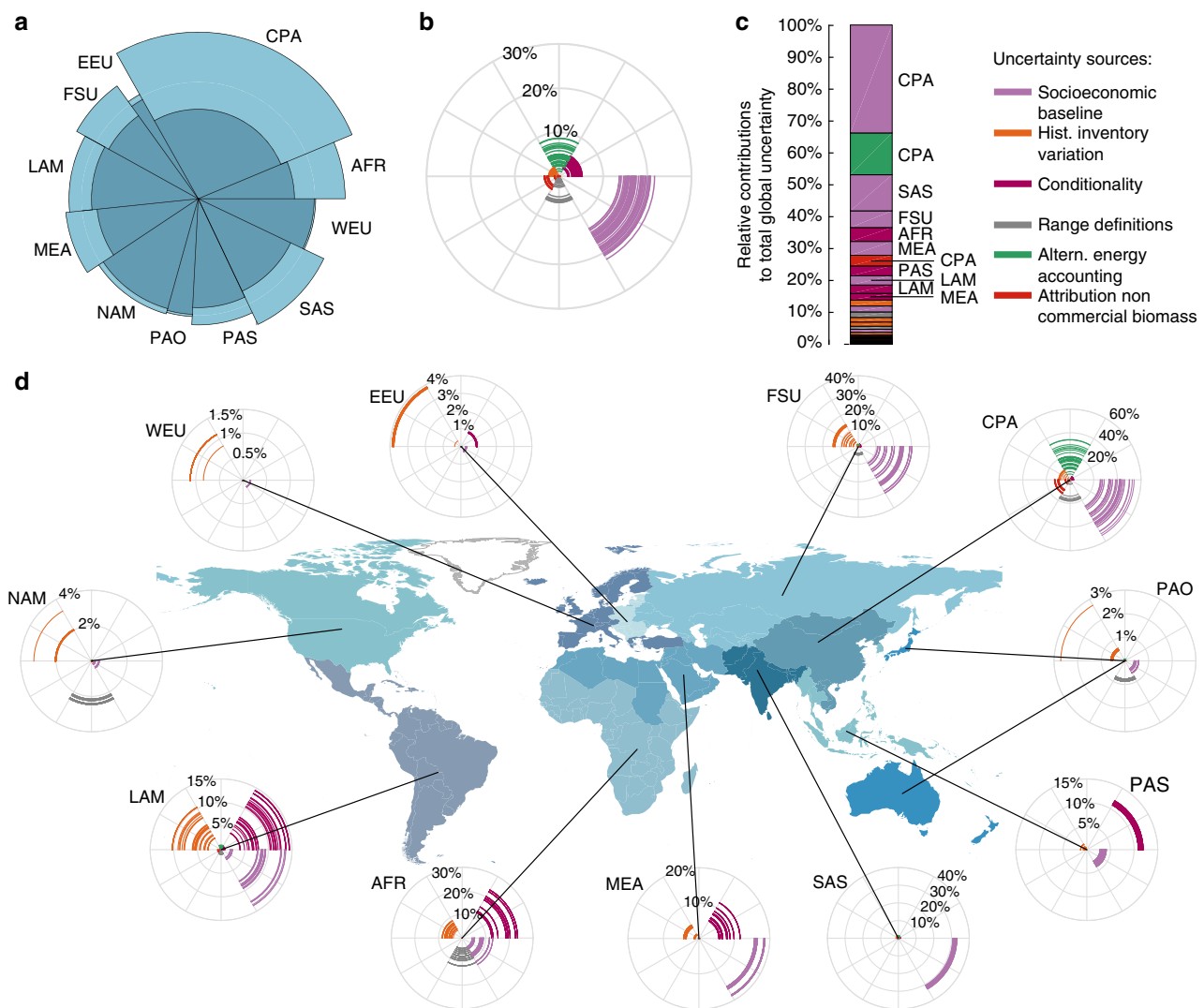

**Figure 3 | Regional contributions of uncertainty sources to overall NDC emission projection uncertainty.** (**a**) Regional emissions contributions to global emissions and uncertainty under the full implementation of current NDCs. Shadings show the minimum–maximum range of emissions estimates per region; (**b**) Estimates of the magnitude of uncertainty induced in 2030 per source relative to the median estimate; (**c**) Average contribution to full uncertainty range in 2030 per uncertainty source with the 10 most important contributions identified by region; (**d**) As **b** but per geographical region. AFR, Sub-Saharan Africa; CPA, Centrally Planned Asia and China; EEU, Central and Eastern Europe; FSU, Former Soviet Union; LAM, Latin America and the Caribbean; MEA, Middle East and North Africa; NAM, North America; PAS, Pacific OECD; SAS, South Asia; PAS, Other Pacific Asia; WEU, Western Europe. Country borders use the simplified TM World borders, provided by Bjorn Sandvik (thematicmapping.org).

New Zealand), show much smaller variations. Uncertainties are thus larger in the developing than in the developed world, with the exception of FSU, where uncertainties are also large due to variations between the historical data sets we use in this study.

In four regions, the socioeconomic baseline uncertainty remains the dominant uncertainty, also at the regional level (CPA, SAS, MEA and FSU). In most cases this is because countries in these regions have specified NDC targets in terms of carbon intensity improvements (the amount of $CO_2$ emitted per unit of economic output), which thus can lead to either higher or lower absolute emissions in 2030, depending on the socio-economic development of these regions over the coming decade. China and India have such targets, in combination with renewable targets which are also influenced by the socioeconomic development (Supplementary Data 1). In some cases, for example for the Russian Federation, NDC targets are so weak that reaching them under certain socioeconomic assumptions would imply emissions to raise above baseline emissions levels. Variations in socioeconomic baselines, affecting energy demand and

technological change[14], in this case thus also result in variations in absolute emissions for this region.

Under the Paris Agreement, countries can on a voluntary basis make use of what is referred to as internationally transferable mitigation outcomes[1] (or emission credits). However, when doing so, 'Parties shall [...] promote sustainable development and ensure environmental integrity and transparency, including in governance, and shall apply robust accounting to ensure, inter alia, the avoidance of double counting'[1]. Guidance on the precise accounting of these transferable outcomes is currently still being developed. However, the explicit requirement for Parties to promote environmental integrity when using such credits suggests that credits achieved through overly unambitious domestic targets are likely to receive pushback. In our analysis, we do not include the use of transferable outcomes between regions, but if admitted and used they could increase global emissions by 0.2–0.5 $GtCO_2e\ yr^{-1}$ in 2030 in SSP2. Regions with emission targets set above their no-climate-policy baseline levels can still see slight increases in emissions due to system-wide

effects of global mitigation action. The latter variation is smaller than 2% in our analysis.

In other regions, the conditionality of NDC targets introduces the largest uncertainty. Countries can specify their NDCs conditional on, for example, the availability of international finance or technological support. Uncertainty about whether these conditions will be met introduces uncertainties of about 10–20% in four world regions (AFR, PAS, MEA and LAM). Furthermore, particularly in Centrally Planned Asia (CPA), the unspecified choice of energy-equivalence method for renewable energy expansion targets in China introduces a large uncertainty of up to ~30%.

Also the imprecision of historical emission inventories introduces uncertainties in NDC estimates of up to about 10% in developing country regions (LAM and AFR), which have been subject to less frequent emission reporting in the past, and in economies in transition (FSU). Variations in historical emission inventories change the starting points of reference baselines relative to which reductions are measured. For FSU, one of the three historical emission datasets we use in this analysis reports 14–16% lower emission in 1990 compared to the two other data sets. Variations in historical emission inventories also affect other regions in which many countries specify their NDC as reductions from a fixed historical base year (like NAM, WEU, EEU and PAO). However, the relative variation of this contribution is small, and never exceeds 4%. These uncertainties can easily be reduced by agreeing that UNFCCC inventory data be used for the NDCs. However, as also these can be updated and adjusted, this would not entirely remove the general uncertainty in emission estimates, which can be around 10% globally[16].

Finally, we find that the uncertainty related to whether non-commercial biomass is counted towards national renewable energy targets has only a small effect. This is even the case in regions like South Asia or Sub-Saharan Africa, which today still heavily rely on this energy source. The reason for this is that renewable energy targets from the NDCs of these regions are often explicitly expressed as a renewable share of electricity generation or installed capacity of a certain source, and thus already eliminate the inclusion of non-commercial biomass in their accounting (see Methods).

**Post-2030 implications of NDC range.** Uncertainty in projected year-2030 emissions has important implications for the achievement of the Paris Agreement's long-term temperature goal. To begin with, the various NDC interpretations represent very different levels of global mitigation effort. This is illustrated in Fig. 4a, where the implied global carbon prices for achieving the emissions reductions described by the NDCs are shown. These implied global carbon prices range from 3 to 26 USD $tCO_2^{-1}$ in 2030. Globally our NDC assessment thus results in lower emissions compared to what is estimated for a world in absence of climate policies. However, this is only under the assumption that no credits that are achieved by setting weak targets above baseline levels are traded between regions (see earlier). Overall, the potential range of 2030 carbon prices implied by the NDCs is well below the range for 2 °C-consistent pathways, as assessed by the Intergovernmental Panel on Climate Change (IPCC). For comparable metrics, the IPCC[17] assessment suggests an interquartile range of ~35–70 USD $tCO_2^{-1}$.

The range of implied carbon prices by the NDCs also has important further consequences. The Paris Agreement specifies that the efforts of all Parties will represent a progression over time[1] and subsequent NDCs are thus supposed to become increasingly stringent. Understanding what the current starting point for this progression is, and what metric would be most

appropriate to measure it[18], is an important question and subject to on-going discussions within the UNFCCC.

Earlier studies have reported important tradeoffs between near-term action and the achievement of stringent mitigation goals in the long term[19–23]. Here we quantify how these tradeoffs play out for the current NDCs. A global cost-optimal pathway in our modelling framework (see Methods) applies a globally uniform carbon price which rises over time with the discount rate. Doing less before and by 2030 (reflected by low implied pre-2030 carbon prices) requires doing more afterwards. This is illustrated in Fig. 4b, where the relationship between 2030 GHG emissions and pre-2030 carbon prices is shown (thin black line). Simultaneously, the tradeoff for post-2030 carbon prices is shown when aiming to limit global mean temperature increase to below 2 °C with about 66% probability relative to preindustrial levels (dashed line, henceforth referred to as limiting warming to below 2 °C) or to return warming to below 1.5 °C by 2100 with more than 50% probability relative to preindustrial levels (dash-dotted line, henceforth limiting warming to 1.5 °C). Starting from the potential range of 2030 GHG emission outcomes under the NDCs, the post-2030 carbon prices required to limit warming to below 2 °C range from 57 to 84 USD $tCO_2^{-1}$ in a middle-of-the-road SSP2 world (Fig. 4b). For the sake of consistency between a 2 °C temperature goal and the near-term actions described in the NDCs, such a sudden increase in carbon prices in 2030 (by a factor of about four to more than 25) is to be anticipated and planned for. Finally, we found no pathways which comply with limiting warming to 1.5 °C by the end of the century from 2030 emissions consistent with current NDCs in a middle-of-the-road (SSP2) world. Only when GHG emissions remain below about 44 $GtCO_2e$ yr$^{-1}$ in 2030 (42 $GtCO_2e$ yr$^{-1}$ with GWP-100 values from ref. 9), our model can still provide such pathways. This maximum level is slightly higher than the 37–40 $GtCO_2e$ yr$^{-1}$ range identified in an earlier UNEP report[24].

These results and their related challenges differ depending on the socioeconomic development one assumes for the coming decades and further until the end of the century. For instance, in a global society that is characterized by profound international cooperation with an emphasis on green growth (SSP1) the NDCs would by 2030 result in implied carbon prices in the range of 3–11 USD $tCO_2^{-1}$ and this would keep both the option to limit warming to below 2 and 1.5 °C open (Supplementary Fig. 1). However, in a world characterized by regional rivalry and resurgent nationalism, as assumed in the SSP3 storyline, year 2030 implied carbon prices range from 9–26 USD $tCO_2^{-1}$ but still foreclose limiting warming to either 1.5 or 2 °C (Supplementary Fig. 2).

Ensuring that GHG emissions end up at the lower end of the NDC range would thus limit the risk of temperature targets becoming unattainable[19–23]. Furthermore, earlier research has shown that near-term emissions reductions reduce the reliance on uncertain carbon-dioxide removal technologies that are typically assumed to be available in 1.5 and 2 °C consistent scenarios[17,21]. Lowering near-term emissions beyond the current NDCs would also provide a more consistent signal to industry and society by reducing the necessity for sudden increases in carbon price around the year 2030.

Increasing the stringency of climate action in the next round of NDCs could eliminate this price jump altogether. If NDCs manage to bring down year-2030 emissions to about 40 $GtCO_2e$ yr$^{-1}$, the price jump between implied pre- and post-2030 carbon prices would disappear for limiting warming to below 2 °C (for the middle-of-the-road scenario assumptions of SSP2, Fig. 4b). For returning warming to below 1.5 °C by 2100, that level is lower, at around 32 $GtCO_2e$ yr$^{-1}$ in 2030. Similar tradeoffs can be found for other metrics, for example for

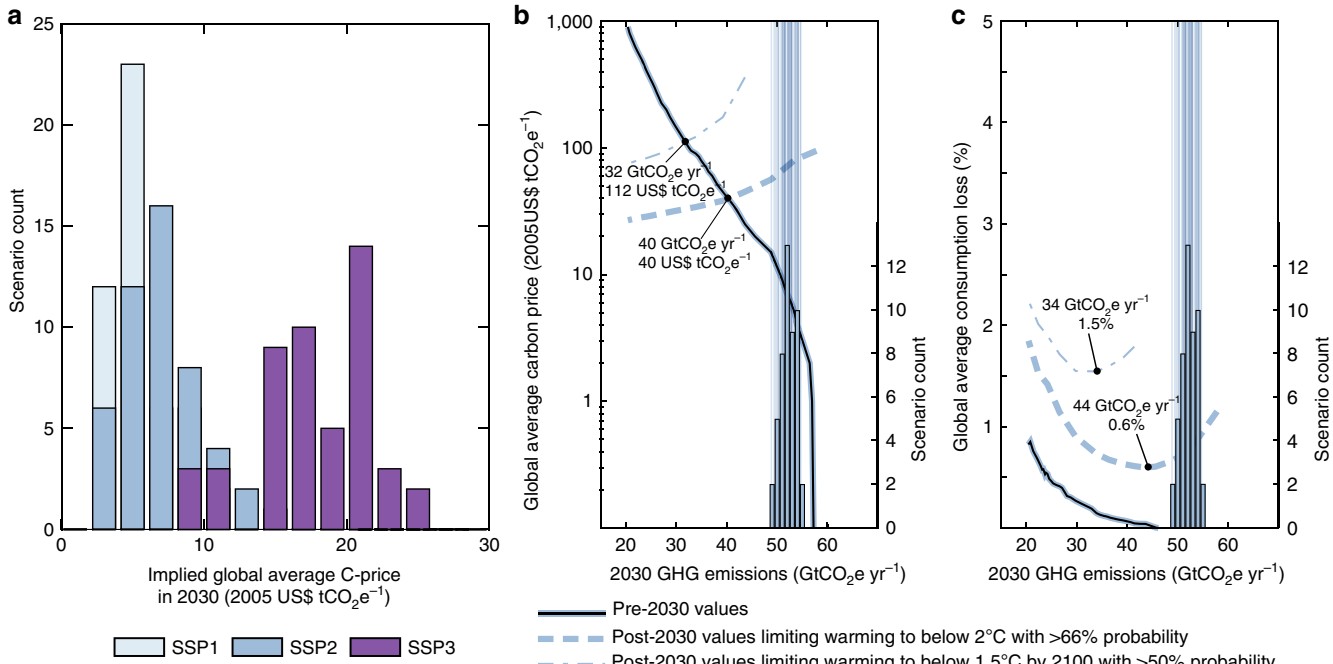

**Figure 4 | Tradeoffs between 2030 NDCs and long-term temperature goals of the Paris Agreement.** (**a**) Global average carbon prices implied by the NDCs grouped by underlying socioeconomic development (SSP1, SSP2 and SSP3); (**b,c**) Tradeoffs between pre-2030 costs (solid line; global average carbon prices in **b**, global average consumption losses in **c**; see Methods for technical descriptions) and post-2030 costs in line with limiting warming to below 2 °C (dashed lines) and limiting warming to below 1.5 °C by 2100 (dash-dotted line) for a world with a middle-of-the-road socioeconomic development (SSP2, ref. 14 for background). The histogram and vertical lines show the distribution of SSP2 NDC estimates from this study (scenario count for the histograms is shown on the right axis).

economy-wide mitigation costs like consumption losses or the speed of upscaling of renewable energy infrastructure (Supplementary Fig. 3). Post-2030 mitigation costs for limiting warming to below 2 °C would be minimized in a middle-of-the-road SSP2 world if 2030 emissions are limited to about 44 $GtCO_2e \ yr^{-1}$, and to about 34 $GtCO_2e \ yr^{-1}$ for 1.5 °C (Fig. 4c). The exact values vary with the assumptions of the underlying socioeconomic development (Supplementary Figs 1 and 2). Finally, it is important to note that these consumption loss estimates neither account for economic benefits of avoiding climate impacts nor account for health, mobility or other co-benefits of climate change mitigation.

## Discussion

The results above show the potential impact and consequences of some of the ambiguities which are present in the current NDCs. Our analysis focuses on uncertainties related to the energy system, but the full uncertainty is likely even larger. There are some uncertainties which we have explored as a sensitivity case only. For example, there is uncertainty about the degree to which climate policies are included in the reference scenario of countries' NDCs. If a country does not provide a reference scenario, we assumed their NDC target to be expressed relative to a no climate policy baseline. These baselines vary in our analysis with the overall socioeconomic assumptions of the different SSPs. However, when we instead assume that country pledges made under the Cancún Agreement[25] are already included in the reference scenario for these countries, our estimates for 2030 would be about 0-0.4 $GtCO_2e \ yr^{-1}$ lower. A further uncertainty and limitation of our analysis is that our modelling framework operates on a decadal time step, and NDC targets are assigned to the nearest step. However, when extrapolating trends implied by

the NDCs from 2025 to 2030 our estimates for the year 2030 are not consistently lowered at a global scale (Supplementary Note 1). Also the choice of GHG equivalence metric adds to the uncertainty. Here, our analysis has been carried out based on the GWP-100 metric values from the IPCC AR4 (ref. 10), although assuming other metrics during the assessment will further influence the estimates (Table 2 and Supplementary Table 1).

Finally, the accounting of land-use-related mitigation has been identified earlier as an important source of uncertainty[3,26]. We do not explicitly explore uncertainties related to this sector, but show with a simple comparison of earlier published land-use uncertainties related to NDCs[26] that in some regions, particularly Sub-Saharan Africa (AFR), Latin-America (LAM) and Pacific Asia (PAS), the uncertainty of land-use emissions might play an important role (Supplementary Fig. 4). Land-use emissions are currently of the order of 4 $GtCO_2 \ yr^{-1}$, and come with high inter-annual variability and uncertainties[27]. Moreover, the large anthropogenic contributions of peat drainage and burning are often excluded from land-use emissions estimates[28]. Fully including land-use emissions would be an important avenue for further research and assessment.

Having quantified key contributing factors to the global and regional uncertainty surrounding emission projections for NDCs, the question arises whether areas for improvement can be identified. Not all uncertainties are created equal[29]. Some are the result of technical imprecisions in the formulation of the NDC, for example, the assumed methodology of renewable energy accounting or the attribution of non-commercial biomass. These uncertainties (of up to ~10% globally) could be reduced by simple clarifications. Also uncertainties related to historical inventories can be reduced in this way by agreeing on robust and transparent rules.

Other uncertainties are harder to reduce, because they are the result of politically valid choices that would be unaffected by technical clarifications. The clearest examples of these are targets expressed as ranges or the specification of climate actions as intensity improvements. There is ambiguity (or scenario uncertainty[29]) in the economic development that underlies these intensity target uncertainties and they thus appear to be irreducible to a large degree for as long as countries choose to express their actions in this way. Some uncertainties, like the conditions attached to particular NDC actions, are in both categories. On the one hand, they are a valid political choice. On the other hand, they can also be reduced by improving clarity of whether and when conditions for implementing certain actions are met. The latter is not just a task for the developing countries that often include such conditional targets. It is equally a task for more developed regions to provide greater clarity about the future availability of funding and other types of support. Altogether, these actions can facilitate an effective ratcheting up of efforts in 2018 and beyond in the context of an effective implementation of the Paris Agreement.

## Methods

**Modelling framework.** We use the Integrated Assessment Modelling framework of the International Institute for Applied Systems Analysis (IIASA IAM[14]) for estimating the global and regional GHG emission consequences of the current NDCs. At its core, the IIASA IAM contains a linear programming energy-system optimisation model called MESSAGE[30,31], coupled to a macroeconomic model and a land-use model[32,33] emulator. For more detailed information about the modelling framework see the online model documentation at http://data.ene.iiasa.ac.at/message-globiom/, and ref. 14 for a description of the SSP implementation in the modelling framework.

**Analysis structure and uncertainty assessment.** To explore the influence of the six uncertainty dimensions listed in Table 1, a factorial scenario design has been implemented which includes all potential combinations between uncertainty dimension interpretations. Figure 1 illustrates the scenario design resulting in 144 unique sets of NDC interpretations. The overall modelling structure and approach is illustrated in Fig. 5. The influence of one specific uncertainty dimension is defined as the difference between scenario pairs which are precisely the same in all dimensions but one. For example, the influence of dimension α (for example historical emissions variations) is computed as

$$\Delta_\alpha = \left\{ \left| E_{\alpha_i, \beta, \gamma, \delta, \varepsilon, \zeta} - E_{\alpha_j, \beta, \gamma, \delta, \varepsilon, \zeta} \right| \; \middle| \; i, j = \{1 \ldots n_{var}\} \; \& \; \forall [\beta, \gamma, \delta, \varepsilon, \zeta] \right\},$$

in which $\Delta_\alpha$ is the set of differences between emissions ($E$) in 2030 of scenario pairs that have exactly the same assumptions in dimensions $\beta$, $\gamma$, $\delta$, $\varepsilon$ and $\zeta$, but differ in their assumption for uncertainty dimension $\alpha$, and this for all available variations ($n_{var}$) in $\alpha$. In two out of six cases $n_{var}$ equals three and in four out of six cases it equals two.

**Estimation of post-2030 implications.** Implied pre-2030 carbon prices have been derived by estimating the global average carbon price required in 2030 to achieve the estimated emission reductions under various interpretations of the NDCs in a cost effective manner. Post-2030 carbon prices consistent with limiting global mean temperature increase to below 2 °C relative to preindustrial levels with at least 66% probability or returning warming to below 1.5 °C by 2100 with at least 50% probability (allowing temperature to temporarily exceed the 1.5 °C threshold) have been derived by a two-stage modelling approach. In the first stage, the model is run myopically until 2020 and 2030 with the aim to represent a specific emission reduction. In a second stage, the energy-system development (including its emissions) is frozen until 2030 and the model then optimizes the energy-system over the entire twenty-first century, with the aim to stay within a cumulative GHG emissions constraint (about 820 GtCO₂e for 1.5 °C, and ∼1,890 GtCO₂e for 2 °C, both from 2010 to 2100 in SSP2). Pre- and post-2030 carbon prices reported in Fig. 4 are then derived by the implied 2030 carbon prices from the myopic step and the full century optimization, respectively. Global average consumption losses are estimated by comparing the consumption of the full century optimisation for limiting warming to below 1.5 or 2 °C with the consumption estimated in the no climate policy baseline. Consumption losses neither account for the economic benefits of avoiding climate impacts nor account for health, mobility or other co-benefits of climate change mitigation. Pre-2030 consumption losses are computed over the 2020–2030 period, post-2030 losses over the 2030–2050 period. All monetary values are provided in 2005 USD and all losses are discounted back to a common 2020 base year (discount rate: 5%).

**Emissions and climate.** Climate outcomes have been computed with the MAGICC reduced-complexity carbon-cycle and climate model[34,35] in a probabilistic setup[36–38]. GHG emissions refer to emissions of the gases from the Kyoto-GHG basket[39], which here includes CO₂, CH₄, N₂O, HFCs, PFCs, and SF₆. NF₃, a gas recently added to this basket[40], is not yet included in these estimates but has a very minor global contribution. In this study, we express CO₂ equivalence by means of GWP-100 values from ref. 10.

**Implementation of NDCs.** Mitigation actions specified in the NDCs (available at: http://www4.unfccc.int/submissions/INDC/) are identified at the national level and subsequently aggregated and analysed at the level of eleven world regions (Supplementary Table 2). For the year 2020, we assume that country pledges under the Cancún Agreement[25] are fully implemented. The following sections document the approach followed for historical emissions, emissions projections, and the interpretation and implementation of (intended) NDCs in our modelling framework.

**Historical data.** *Deriving and extending national historical GHG emissions.* Historic emission levels ($GHG_{iso}$ in MtCO₂e yr⁻¹, using AR4 GWP-100 values for the conversion to CO₂-equivalent), for individual countries ($iso$), were derived by adding CO₂ emissions excluding short-cycle biomass burning ($CO_{2,iso}$ in TgCO₂ yr⁻¹), CH₄ emissions ($CH_{4,iso}$ in TgCH₄ yr⁻¹), N₂O emissions ($N_2O_{iso}$ in Tg N₂O yr⁻¹) and F-gases ($FGases_{iso}$ in Tg CO₂e yr⁻¹ including HFCs, PFC and SF₆). In order to account for CO₂ emissions from biomass burning ($CO_2bb_{iso}$ in TgCO₂ yr⁻¹), national level data from the AR5 database was also added.

$$GHG_{iso} = \frac{(CO_{2,iso} + CH_{4,iso}*25 + N_2O_{iso}*298 + FGases_{iso} + CO_2bb_{iso})}{1,000}.$$

*Scaling of national historical emissions.* Historical national emission levels are used by many countries as a reference point relative to which future emission reduction goals are defined. Therefore, in order to ensure consistency between future national emission targets which are aggregated and imposed on model specific regions, any inconsistencies between historic national emissions and historic model emissions need to be addressed.

Historical emissions in the IIASA IAM have been calibrated to various different sources, depending on the gases. Differences between aggregate national historical emissions and the model are hence to be expected. Emission data are not available on the national scale for all the sources used in the calibration of the model. National level emissions are rescaled to match the respective model region emissions, following a method based on ref. 41. In a first step, the difference in emissions ($EDiff_{r,t}$) for region ($r$) in time-step ($t$) is derived by subtracting the sum of the emissions for countries ($GHG_{iso;t}$) within a given region ($reg$), from the GHG emissions of the model region ($GHG_{r,t}$).

$$EDiff_{r;t} = GHG_{r;t} - \sum_{iso}^{reg} GHG_{iso;t}.$$

The emission difference for a given region is then distributed across countries within that region, based on the countries emissions in proportion to the sum of the national GHG emissions ($GHGsh_{iso;t}$).

$$GHGsh_{iso;t} = \frac{GHG_{iso;t}}{\sum_{iso}^{reg} GHG_{iso;t}},$$

$$GHG_{iso;t;adjusted} = GHG_{iso;t} + EDiff_{r;t}*GHGsh_{iso;t}.$$

This adjustment is kept constant for other historical years.

**Emissions projections.** *Downscaling projected regional GHG developments onto national GHG emission pathways.* Projected national GHG emissions (that is, from 2020 onwards) are derived from regional GHG developments in the respective SSP baseline scenario[14]. Regional emissions are downscaled onto respective countries using the methodology as described in ref. 41. First, the GHG emission intensity ($GHGI$, MtCO₂e GDP⁻¹) development is extrapolated to a chosen convergence year ($CY = 2200$), which lies beyond the time horizon covered by the model (1990–2100), by using the growth rate from the past 10 years of the baseline scenario (2090–2100). The extended time series ($GHGI_r$) is then used to determine a constant annual emission intensity growth rate per country ($GHGIgr_{iso}$) starting from the national emission intensity ($GHGI_{iso}$) in the base year ($BY = 2010$).

$$GHGIgr_{iso} = \left( \frac{GHGI_{r;CY}}{GHGI_{iso;BY}} \right)^{1/(CY-BY)}.$$

If the national emission intensity in the base year is negative, the formula is slightly altered.

$$GHGIgr_{iso} = \left( \left( \frac{GHGI_{r;CY}}{GHGI_{iso;BY}} \right)*-1 \right)^{1/(CY-BY)}.$$

Baseline national emission intensities ($GHGI^*_{iso;t}$) are subsequently determined by multiplying the national emission intensity of the previous time step with the

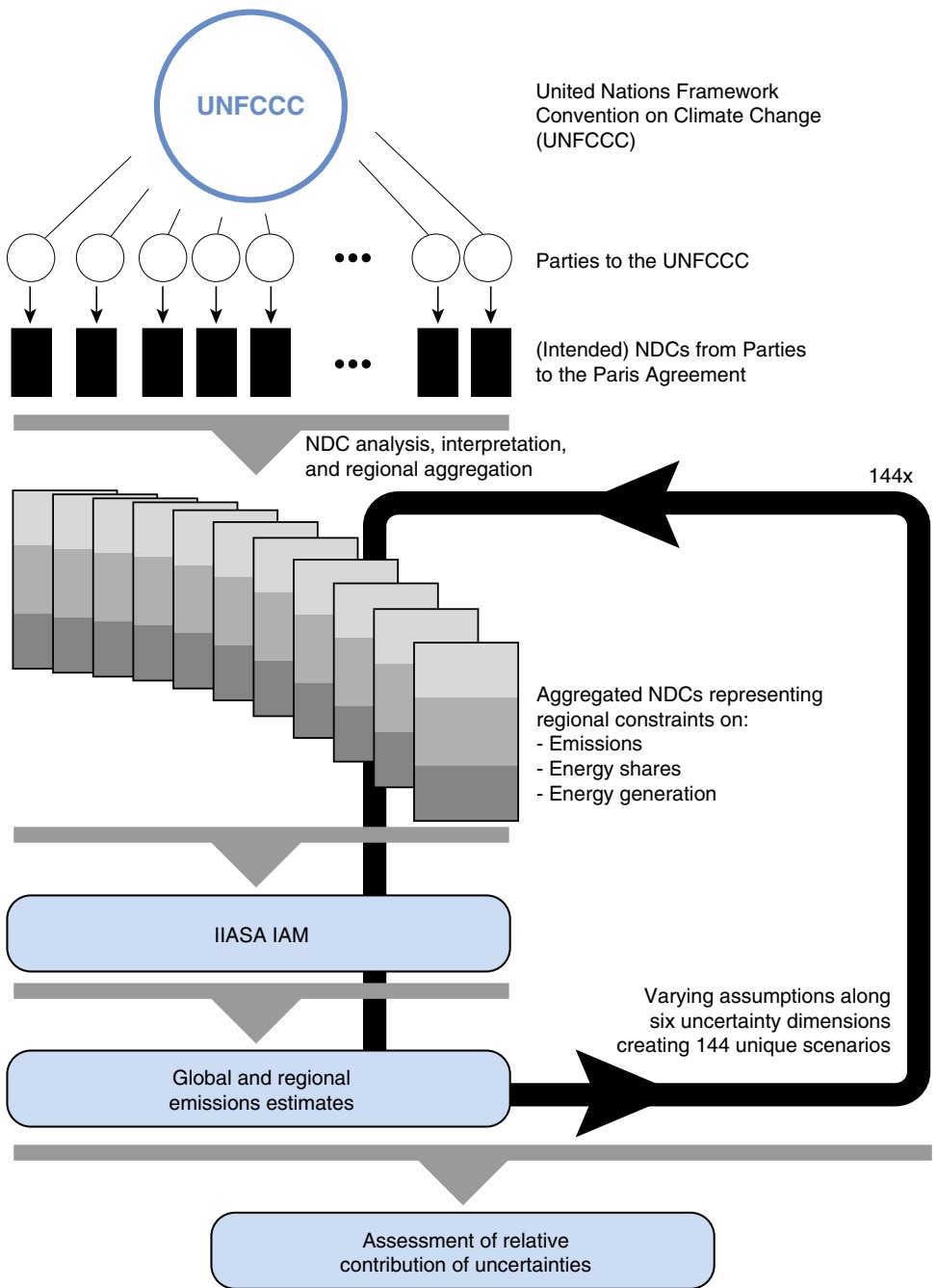

**Figure 5 | Schematic representation of the analysis of NDC uncertainties.** Emissions estimates for a total of 144 scenarios are computed and ultimately compared in a pair-wise fashion. (Intended) NDCs of Parties to the Paris Agreement are available at: http://www4.unfccc.int/submissions/INDC/. Documentation of the IIASA IAM is available at: http://data.ene.iiasa.ac.at/message-globiom/.

constant national emission growth rate:

$$GHGI^*_{iso;t} = GHGI_{iso;t-1} * GHGIgr_{iso}.$$

Consistency between aggregate national emission levels and regional baseline emissions is ensured by once again applying the scaling approach as described earlier.

**Interpretation and implementation of NDCs.** *General approach.* NDCs are provided by individual countries and distinguish between conditional and unconditional targets and/or policies. Our modelling framework operates on a coarser geographical resolution (Supplementary Table 2), and national pledges and targets must thus be translated from the national level into model region constraints. However, our framework can also provide approximate estimations of single country NDCs (Supplementary Note 2, and Supplementary Fig. 5).

Three general types of constraints are imposed within the IIASA IAM based on the NDCs: (1) Emission constraints, (2) Share constraints, and (3) Generation constraints.

All of these constraints are first derived on the national level and subsequently aggregated to the respective model regions. Unquantified targets or non-emission targets relating to the land-use sector were not taken into account in this analysis. If data required for the calculation of targets, such as historical emissions or GDP data used in the downscaling process is not available, then the countries are omitted. For example, this was the case for some small island states like the Seychelles, Grenada, Dominica or other territories like Greenland. If country data are missing in one of the historic emission data sets, it is replaced by data from one of the other two data sets. The default choice here is EDGAR[42] (see below). NDC estimates for 2030 start from an implementation of the conditional Cancún Agreement targets, and both are based on one consistent historic data set.

**Emissions constraints.** 'Emission constraints' are provided in three different forms most of which are expressed relative to either a national reference emissions trajectory (often referred to as business as usual or BAU) or national historic emission levels: (1) absolute reduction targets, that is, a fixed quantity of emissions to be reduced; (2) a percentage reduction, that is, a relative share of emissions to be reduced; and (3) intensity reductions, for example, emissions per GDP or per capita to be reduced by a certain percentage.

Targets that are expressed relative to a historical base year are computed relative to the harmonized emission inventory data (see earlier). When BAU emissions are not specified by the NDC, national downscaled emissions from the regional no-policy reference baseline were used. Variations have been implemented to account for evolutions in line with SSP1, SSP2 and SSP3, as well as a sensitivity case which includes the Cancún Agreement pledges until 2020 but resumes a baseline development afterwards. Furthermore, unless specified otherwise, all emission reduction targets are assumed to apply to all sectors and to relate to all gases (that is, total GHG emissions). For each country, the maximum allowable emissions were calculated. If no target was specified, downscaled BAU levels were applied. Subsequently, these were imposed as upper limits on GHG emissions for each of the 11 regions, respectively (Supplementary Table 3).

**Share constraints.** 'Share constraints' refer to NDC targets which aim to provide a certain share of a specific energy level through a specific set of energy forms. The different types are: (1) renewable energy forms as a share of primary energy; (2) non-fossil energy forms as a share of primary energy; (3) renewable energy forms as a share of electricity generation; (4) non-fossil energy forms as a share of electricity generation; and (5) renewable energy forms as a share of final energy. The general formula used to calculate the regional share is as follows:

$$shr_{r;TY} = \sum_{iso}^{reg} \frac{Energy_{iso;RY}}{Energy_{reg;RY}} * Tshr_{iso;TY},$$

where $shr_{r;TY}$ is the minimum share of either renewables or non-fossil energy in a specific region ($r$) for the target year ($TY$); $Energy_{iso;RY}$ is the total national energy for a given energy form (primary energy, electricity generation or final energy) in the reference year ($RY$); $Energy_{reg;RY}$ is the total regional energy for a given energy form (primary energy, electricity generation or final energy) for a given region ($reg$) in the reference year ($RY$); and $Tshr_{iso;TY}$ is the national target share in the target year ($TY$) as defined by the NDC of each country. The target year is defined by the NDC of each country, which in most cases is 2030. The reference year, in this case 2010, is the historical year from which energy data are used. We use historical data as there is no energy data available at the national level in the IIASA IAM.

We harmonize the different types of share constraints of the constituting countries in each respective region. For example, for the Sub-Saharan Africa model region, Ghana states in its NDC that it aims to provide a certain share of primary energy through renewables. At the same time, Namibia's NDC aims to provide a certain share of its electricity generation though renewables. Technically, the model would not consider these targets to be cumulative at the regional level if they were implemented separately, resulting in an underachievement of the NDCs from a regional perspective. Therefore, for each region, these NDC targets are converted to a dominant target type of either the largest country in terms of energy share or of the majority of the countries within a specific region.

In such cases, the above formula is reformulated as follows:

$$shr_{r;TY} = \sum_{iso}^{reg} \frac{Energy_{iso;RY}}{Energy_{reg;RY}} * NewTshr_{iso;TY}.$$

$NewTshr_{iso;TY}$ is the recalculated national target share in the target year ($TY$) translated from its original type to the dominant type within a region. National NDC shares are translated in regional shares by applying the formulas provided below.

National NDC shares defined as 'renewables as a share of primary energy' are translated to regional shares defined as 'renewables as a share of electricity generation' with the following formula for the direct equivalent accounting method:

$$NewTshr_{iso;TY} = \frac{Tshr_{iso;TY} * \left(\frac{RePeElec_{iso;RY}}{RePe_{iso;RY}}\right) * PeTot_{iso;RY}}{ElecTot_{iso;RY}}.$$

$RePeElec$ is the electricity produced by renewable energy sources in primary energy equivalent; $RePe$ is the total primary energy produced from renewable energy sources; $PeTot$ is total primary energy; and $ElecTot$ is total electricity. For the substitution accounting method, the following formula is applied,

$$NewTshr_{iso;TY} = \frac{Tshr_{iso;TY} * \left(\frac{RePeElec_{iso;RY}}{RePe_{iso;RY}}\right) * PeTot_{iso;RY} * 0.38}{ElecTot_{iso;RY}}.$$

National NDC shares defined as 'renewables as a share of electricity generation' are translated into regional shares defined as 'renewables as a share of primary energy' with the following formula for the direct equivalent accounting method:

$$NewTshr_{iso;TY} = \frac{Tshr_{iso;TY} * ElecTot_{iso;RY}}{PeTot_{iso;RY}}.$$

For the substitution accounting method, the following formula is applied in this case:

$$NewTshr_{iso;TY} = \frac{Tshr_{iso;TY} * ElecTot_{iso;RY} * \left(\frac{1}{0.38}\right)}{PeTot_{iso;RY}}.$$

National NDC shares defined as 'renewables as a share of electricity generation' are translated into regional shares defined as 'renewables as a share of final energy' with the following formula for the direct equivalent accounting method:

$$NewTshr_{iso;TY} = \frac{Tshr_{iso;TY} * FeElec_{iso;RY}}{FeTot_{iso;RY}}.$$

$FeElec$ is the electricity at the final energy level, and $FeTot$ the total final energy.

National NDC shares defined as 'renewables as a share of electricity generation' are translated into regional shares defined as 'non-fossils as a share of primary energy' with the following formula for the direct equivalent accounting method:

$$NewTshr_{iso;TY} = \frac{Tshr_{iso;TY} * ElecTot_{iso;RY} * \left(\frac{NFElec}{ReElec}\right)}{PeTot_{iso;RY}}.$$

$NFElec$ is the electricity produced by non-fossil energy forms, and $ReElec$ the electricity produced by renewable energy forms. For the substitution accounting method, the following formula is applied in this case:

$$NewTshr_{iso;TY} = \frac{Tshr_{iso;TY} * ElecTot_{iso;RY} * \left(\frac{1}{0.38}\right)}{PeTot_{iso;RY}}.$$

*Final regional share constraints.* The IIASA IAM operates at an 11-regional level, and exact future developments of national energy systems are thus not resolved by the model. Hence, it is assumed that the near-term national share of energy remains the same as in the 2010. On the basis of these shares, national targets can be expressed as a share of total regional energy developments. Supplementary Table 4 provides an example of the share constraints for different regions.

The share constraint is formulated as follows:

$$\sum_{r;t} lhs_{r;t} \geq \left( \sum_{r;TY} lhs_{r;t} + \sum_{r;TY} rhs_{r;t} \right) * shr,$$

which can be reformulated as

$$\frac{(1 - shr)}{shr} * \sum_{r;TY} lhs_{r;t} - \sum_{r;TY} rhs_{r;t} \geq 0.$$

$lhs_{r;t}$ represents the energy production from renewable or non-fossil technologies in a given region ($r$) at a specified point in time ($TY$). Energy production from all other energy forms is represented by $rhs_{r;t}$. The variable $shr$ represents the minimum percentage of the total energy production to be provided by either the renewable or non-fossil technologies, depending on the variant of the constraint.

**Generation constraints.** The third constraint type used to implement national targets are those which require specific technologies to generate a pre-specified amount of energy. In fact, most of the countries specify such capacity targets. For example, China aims at upscaling its total installed wind capacity to 200 GW by 2030. For technical reasons, such targets/policies are translated into a bound requiring relevant production technologies to provide a minimum output (GWh) equivalent to the installed capacity. To derive the output based on capacity values, the average efficiency of wind production technologies of the respective region available in the target year is used to translate capacity into output. An example of the regional minimum output requirements for the NDCs are summarized in Supplementary Table 5.

**Data sources.** Both historical developments and future projections are based on published data sources. Baseline scenarios were taken from the IIASA IAM implementation of SSP1, SSP2 and SSP3 as described in ref. 14. Historical GHG emissions use data from PRIMAPHIST[11], CAIT[43] and EDGAR[42] with F-gases from ref. 44. Historic national GDP (PPP) and population projections were taken from ref. 45, while future national GDP (PPP) and national population developments are taken from refs 15 and (ref. 46), respectively. NDC targets are assessed as 3 September 2017 (ref. 47). National Cancún pledges are based on ref. 48, and energy data on ref. 49. Initial assessments of the NDCs were based on the INDC & NDC fact sheets by R. Alexander and M. Meinshausen from the Australian-German Climate and Energy College of the University of Melbourne, available at: http://climate-energy-college.org/ndc-indc-factsheets.

**Code availability.** The current code base of the IIASA IAM, developed over more than two decades at the International Institute for Applied Systems Analysis (IIASA), is not available in a publicly shareable version. Future model versions which are currently under development will be shareable and under an open source license. The code will continue to be developed and hosted by IIASA's Energy Program (ENE; http://www.iiasa.ac.at/web/home/research/researchPrograms/

Energy/MESSAGE.en.html). Requests for code should be addressed to the ENE Program contact.

**Data availability.** The data that support the findings of this study are available from the corresponding author upon reasonable request.

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

## Acknowledgements

J.R., O.F., V.K., J.J.J.Z. and K.R. received funding from the European Union's Horizon 2020 research and innovation programme under grant agreement No. 642147 (CD-LINKS). M.M. is supported by an Australian Research Council (ARC) Future Fellowship (grant number FT130100809).

## Author contributions

J.R., M.M. and K.R. designed the research. O.F. led the implementation and documentation of the NDCs in the IIASA IAM, with inputs from J.J.J.Z. and all other authors. V.K. prepared the data for assessing energy accounting ambiguities. J.R. carried out the analysis of the modelling results, led the writing of the paper and created all figures. All authors contributed to the discussion and interpretation of the results.

**Additional information**

**Competing interests:** The authors declare no competing financial interests.

**Publisher's note**: 

