## [Peer Review File · Nature Communications]

Editorial Note: Parts of this peer review file have been redacted as indicated to remove third-party material where no permission to publish could be obtained

Reviewers' comments:

Reviewer #1 (Remarks to the Author):

The study by Rogelj et al. deals with an interesting topic. At the heart of the study is a decomposition and regional attribution of the sources of uncertainty in NDC emissions estimates. But it is not very clear how this paper contributes to current knowledge. There are several papers in this domain from modeling teams and this one doesn't seem to add any value, because it is not clear how the model used in this paper is justified to answer the question of interest (or "provide new science", as the authors claim) or whether the authors have undertaken the task of actually representing real-world policies that will be put in place to achieve the NDCs. Unless a model with sufficient regional disaggregation is used and sufficient representation of real-world policies is undertaken, this paper is not good enough to be published.

The main issue I have with the paper is that the IAM used in the analysis has very high levels of regional aggregation; for example, major players such as U.S., China and India are not even regions of their own. For instance, U.S. and Canada are clubbed into one region in the model. This raises several questions. To what extent do these regional aggregations affect the uncertainty ranges? There is huge uncertainty in the NDC definitions of some major countries. For instance, did the authors assume that the Clean Power Plan is in place in the U.S. (it is still in courts, so what was assumed?). Another critical example is China. China says their NDC has several components including an emissions intensity target, renewable energy target, peak emissions target, etc. How were these represented? Which of these were chosen and why? Likewise, India's NDC is based on a percentage of cumulative electricity capacity. How can such targets be represented credibly when India is not a region of its own? How are the authors sure that regional definitions do not affect the uncertainty ranges?

It seems that these targets have been converted to emissions levels using some sort of calculations. If so, without explicitly modeling the policies actually in place (for example, CAFÉ standards, Clean Power plan in USA; solar R&D and deployment in India, financing policies in China, etc.), claiming the results to be "new science" seems to be far-fetched. Just take the U.S. for example, there is huge uncertainty in terms of, first, whether the CPP will come to force in the new Presidency, second, even if you assume the CPP will come to force, emissions levels will differ based on whether you do a mass-based CPP or a rate-based CPP and third, whether you assume states can trade emissions credits or not. The IAM community has now for too long relied on economy-wide emissions constraints, and very useful lessons have been drawn from those, including in the IPCC reports. Given all of this rich literature, the need of the hour is papers with at least some reasonable representations of individual policies. For a study such as the present, unless actual policies are modeled with appropriate regional aggregation, the results seem to show a false sense of precision to warrant publication in a high profile journal such as Nature Communications. I would strongly recommend not publishing this paper unless the authors have re-done the modeling of the policies with explicit regional disaggregation of at least half a dozen countries and with explicit modeling of key policies in those regions.

Another concern with this paper is that the figures are too hard to read and make sense. For example, in Figure 1a, what are the individual lines? How many scenarios were analysed in the paper? What's the point being made in Figure 1b in the lines with dots? Figure 2 can easily be converted into a table or may be a simple bar-graph. Figure 3 is not clear at all. Why is there a discussion of carbon prices? If it is to show near-term versus long-term tradeoffs, millions of papers have shown that already. What is important for decision makers is how many nuclear power plants need to be built? What's the rate of change in the energy system transitions (see for example papers by van Vuuren et al. in the Focus Issue on cumulative emissions in Environmental Research Letters. There are also some interesting papers on near-term versus long-term tradeoffs in that Focus Issue). In what way is this "new science"? Also, in Figure 3b, what is shown is 2030 carbon price with respect to 2030 emissions. What can we conclude about near-term versus long-

term from that figure?

In terms of reorganizing figures and tables, I would recommend a major revision of the text and figures that starts describing the results to move from one figure to another to tell a well-knit story, which I think can be done.

A final concern I have is that the authors do not in general seem to make their model available online. I tried my best to find it online and it just does not exist. How are we as scientific researchers to ensure reciprocity? It seems from this paper that the models MESSAGE and MAGICC are black boxes. Surprisingly, there is not even much documentation of the models in the Supplementary. Of course the final decision on this lies with the editor. Nevertheless, I thought I would flag it since this is an open source journal.

Reviewer #2 (Remarks to the Author):

Review Comments on NCOMMS-16-20105-T

Understanding the origin of Paris Agreement emission uncertainties

This paper provides the systematic estimation of emission uncertainty of Paris agreement and its implications for the long-term climate policy. An integrated assessment model is a core of the methodology. Hundreds of scenarios associated with six uncertainty dimensions are produced. The authors show that 47-63 GtCO₂e is the range of 2030's global GHG emissions. This range has critical implications for the feasibility and costs to limit warming to well below 2 or 1.5 °C. The main source of uncertainty to determine 2030's global GHG emissions is socioeconomic assumption where they used Shared Socioeconomic Pathways (SSPs). They finally suggest how to decrease the uncertainty.

The overall text is well written, and the logic is understandable. However, there are several concerns before publishing. Here, I listed some points that could be modified or improved.

1) In the final section, the authors stress that the clarifying socioeconomic assumptions largely contributes to reducing uncertainty. That would be true from this model exercise. However, socioeconomic assumptions are abstract for policy implementation. Looking at the SSPs framework (Riahi, 2016), there are various elements that determine socioeconomic assumptions. Do authors ask to specify all of them in NDCs? I envisage that all countries cannot do such things realistically speaking. Then, I come up with another question whether the authors implicitly mentioned about some of the socioeconomic elements?

2) Following up to the above question, let us assume that socioeconomic assumptions are clarified in NDCs as authors suggest. Then, can we derive the same implications of the long-term mitigation goal addressed in this paper? For example, given the situation where accidentally all countries submit SSP2 conditions in NDCs, GDP, population, energy policies and so on until 2030 exactly follow SSP2 assumption. The uncertainty must be reduced significantly. However, don't we still have a possibility the world goes SSP3 direction which is worst socioeconomic assumptions regarding climate mitigation afterward, and the long-term implication is still uncertain? I may be able to agree that the socioeconomic condition is the main source of INDC emissions in 2030. However, I still feel that there is a gap from this 2030's emissions uncertainty and long-term implication.

3) The methodological description is not sufficient regarding attribution of uncertainty. I could guess that the authors run the all possible combinations of each uncertainty dimension something like 2*3*2*... However, there is no explanation how to attribute the uncertainty from such large number of scenarios. That should be documented elsewhere.

4) The uncertainty regarding land use change emissions seems very important element to understand the uncertainty of future emissions. For example, IPCC AR5 WG3 fig 11.6 presents the range of current emissions is around 5 GtCO₂ and cannot be ignored. Then, how can we interpret the outcome of this study considering such additional uncertainty?

5) How does the regional classification of the integrated assessment model influence to this study? For example, if you have single independent US, China, India, Russia, and Brazil? It would be nice to discuss.

Reviewer #3 (Remarks to the Author):

Overall

The role of adapting to the effects of climate change are important, however, missing from most NDCs. Might be good to make clear that the focus will be rather on mitigation/emissions numbers. You hint at this in the limitations of the study later, but could be good to bring it up front somewhere.

Not always easy to read at points – with dense information. The key messages or what was important from the study therefore doesn't come out so clearly at times. Although I caveat this comment because I don't know the journal nor its audience very well.

Unclear how some uncertainties were calculated which are one of the key parts to this paper. E.g. Page 6 lines 21. 23. This seems to 'jump ahead' a few steps in the logic from INDC assumption to translating into tCO₂e ranges. Would be good to know those steps, perhaps quickly in a flow/chart to explain your steps.

How do the pathways to 2020, 2030 fit with the overall IPCC expectations? Wasn't clear to me in the discussions: e.g. Page 8, line 20, where you explain that only below 44 Gt in 2030 will mean 1.5C is possible.

Are negative emission scenarios potentially needed for 1.5/2C. I only ask, not to highlight such technological needs (because it is clearly better to have a situation without them), but readers might expect a sentence or paragraph on it.

How are emissions from peat fires accounted in the model, or other non-sovereign emissions from aviation/shipping? My work on this showed that it is large (e.g. >1 Gt p.a.) and difficult to measure. You hint at this in paragraph on p10 line 9.

Specific points

Line 9. 'will be taking place' ◊ by whom? UNFCCC? Clarify

Line 26. 'and their conditions' ◊ Expand as is an important point. I see some in the supplement text but it could also be in the main text. Also dependent on which models and assumptions are applied. It's all pretty subjective which might highlight the differences. Indeed, some analysis takes INDCs at 'face value' rather than making own assumptions on GDP growth of a country (specifically, India). I don't see a discussion of GDP assumptions in the text nor supplement for instance. In Table 1, It would be good to know how GDP assumptions are accounted.

Page 6. Line 21. How is 10-20% uncertainty on international finance provision calculated? Likewise line 23.

We would like to thank the three referees for their remarks and helpful suggestions, which have greatly helped to improve our manuscript. Thanks to the constructive referee remarks we have been able to further clarify our approach and the applicability of our results. In particular, some review comments brought to light that our research question was not clearly communicated. We have therefore made sure that the research question, as well as the scope of our manuscript is now defined much more precisely.

Point-by-point responses to the referee comments are inserted below in blue.

INDIVIDUAL RESPONSES TO THE REFEREES

FIRST REFEREE REPORT

Reviewer #1 (Remarks to the Author):

We thank the reviewer for his/her detailed and constructive suggestions. The reviewer's suggestions have brought to light that important aspects of our analysis could easily be misinterpreted. We have dealt with these issues with great care in our revision. Below, we respond to the various points raised by the reviewer in full detail.

The study by Rogelj et al. deals with an interesting topic. At the heart of the study is a decomposition and regional attribution of the sources of uncertainty in NDC emissions estimates. But it is not very clear how this paper contributes to current knowledge. There are several papers in this domain from modeling teams and this one doesn't seem to add any value, because it is not clear how the model used in this paper is justified to answer the question of interest (or "provide new science", as the authors claim) or whether the authors have undertaken the task of actually representing real-world policies that will be put in place to achieve the NDCs. Unless a model with sufficient regional disaggregation is used and sufficient representation of real-world policies is undertaken, this paper is not good enough to be published.

The main issue I have with the paper is that the IAM used in the analysis has very high levels of regional aggregation; for example, major players such as U.S., China and India are not even regions of their own. For instance, U.S. and Canada are clubbed into one region in the model. This raises several questions. To what extent do these regional aggregations affect the uncertainty ranges? There is huge uncertainty in the NDC definitions of some major countries. For instance, did the authors assume that the Clean Power Plan is in place in the U.S. (it is still in courts, so what was assumed?). Another critical example is China. China says their NDC has several components including an emissions intensity target, renewable energy target, peak emissions target, etc. How were these represented? Which of these were chosen and why? Likewise, India's NDC is based on a percentage of cumulative electricity capacity. How can such targets be represented credibly when India is not a region of its own? How are the authors sure that regional definitions do not affect the uncertainty ranges?

It seems that these targets have been converted to emissions levels using some sort of calculations. If so, without explicitly modeling the policies actually in place (for example, CAFÉ standards, Clean Power plan in USA; solar R&D and deployment in India, financing policies in China, etc.), claiming the results to be "new science" seems to be far-fetched. Just take the U.S. for example, there is huge uncertainty in terms of, first, whether the CPP will come to force in the new Presidency, second, even if you assume the CPP will come to force, emissions levels will differ based on whether you do a mass-based CPP or a rate-based CPP and third, whether you assume states can trade emissions credits or not. The IAM community has now for too long relied on economy-wide emissions constraints, and very useful lessons have been drawn from those, including in the IPCC reports. Given all of this rich literature, the need of the hour is papers with at least some reasonable representations of individual policies. For a study such as the present, unless actual policies are modeled with appropriate regional aggregation, the results seem to show a false sense of precision to warrant publication in a high profile journal

such as Nature Communications. I would strongly recommend not publishing this paper unless the authors have re-done the modeling of the policies with explicit regional disaggregation of at least half a dozen countries and with explicit modeling of key policies in those regions.

RESPONSE:

The reviewer highlights several important issues, which show the need for better clarity about the purpose, strengths and weaknesses as well as the appropriateness of our approach. From the reviewer's feedback, we have identified the following key issues that require clarification:

- (1) The research question this paper is trying to answer, and its novelty.
- (2) What are NDCs? What is modelled? And what is the precision of this?
- (3) The appropriateness of regional aggregation in our modelling approach.

Each issue is clarified separately below.

(1) Research question:

The comments by the reviewer illustrate that there is a misunderstanding as to which research question we are answering with this analysis. The research question we are answering is: How large is the uncertainty in projected emissions under the current formulation of *nationally determined contributions* (NDCs)? And: What are key drivers of this uncertainty?

The comments of the reviewer, however, appear to suggest that our manuscript was understood to answer questions like: Will the NDCs be achieved? How far do current national policies bring us in achieving the NDCs? Can we precisely *predict* where emissions are heading when taking into account the implementation of current policies and measures?

Our manuscript does not intend nor does it pretend to answer any of those. This misinterpretation by the reviewer brought to our attention that we initially made the research question of our manuscript insufficiently clear to the reader, and we have therefore ensured that this is remediated in the revised introduction to our manuscript. By doing so, it should now be clear that our research question does not require modelling of all the aspects listed by the reviewer, like anticipated national policies or sectorial measures not specified in the NDC.

What we do model, on the contrary, is clarified further below in this response.

In a further comment, the reviewer questions the novelty of the presented analysis. He/she indicates that there are several papers published in this domain of modelling and that the way to go is to represent individual policies. We agree with the reviewer that this is one way to go, but it is not the way we have chosen in this study and thus strongly object to the idea that the reviewer's suggestion is the only valid direction of movement. Furthermore, we agree with the reviewer's observation that several publications have been published in the domain of NDC modelling, but strongly disagree with the reviewer's claim that our analysis would not add any value. This is an unsubstantiated and, indeed, unreferenced claim by the reviewer.

The reviewer is correct in pointing out that many studies exist which already estimate potential emissions outcomes of the NDCs. These studies include Fawcett et al. *Science* (2015), Rogelj et al. *Nature* (2016) (and studies referenced therein), Vandyck et al. *Global Environmental Change* (2016), Fujimori et al. *SpringPlus* (2016), as well as the NDC Synthesis Report of the UNFCCC (2016), amongst other studies. We cite many of these in our manuscript. An assessment of the results of these studies shows a large spread in emissions estimates, which remains opaque and

unexplained to date. The studies further hypothesize about the potential drivers of this spread, but without conclusive and/or internally consistent answers.

Our study responds to this, by digging deeper where the limits of the current literature were identified. By identifying how varying assumptions in the interpretation of NDCs determine the spread, we can quantify their contributions. This is done in an integrated global framework which allows for macro-economic interactions between the NDCs. This provides a novel, valuable and extremely timely contribution to the literature. Our assessment regarding the novelty and timeliness of our study is supported by the interactions and feedback we received from policymakers on the potential impact of our study, and the absence of any peer-reviewed study addressing this question to date.

(2) What are NDCs? What is modelled? And what is the precision of this?

NDCs, or nationally determined contributions, are the international pledges made by countries under the UNFCCC. These pledges apply to a variety of issues, including mitigation actions, adaptation actions, means of implementation (climate finance), and other questions, for example related to equity and fairness of contributions. The international pledges captured under the NDCs are thus fundamentally different from policies and measures that are (or will have to be) implemented at the national level, and to which the reviewer is referring to in his/her comments above.

For example, the reviewer rightfully questions how the US NDC will be achieved with or without the Clean Power Plan (CPP), CAFE standards and probably now also with the prospect of a potentially struggling EPA after the US presidential elections. These are generally good questions but do not apply to our research question. They are of importance when one is interested in *how* the pledges of the US NDC will be achieved. However, to understand what flexibility is left open by the US pledge under the UNFCCC they are irrelevant. The intended NDC of the US (which is available via the UNFCCC NDC interim registry page on website: <http://www4.unfccc.int/ndcregistry/>) reads verbatim:

Party: United States of America

Intended nationally determined contribution
The United States intends to achieve an economy-wide target of reducing its greenhouse gas emissions by 26%-28% below its 2005 level in 2025 and to make best efforts to reduce its emissions by 28%.

Since the US ratification of the Paris Agreement, this intended NDC has also become the US' NDC. Ultimately, it will be against this statement that actual 2025 emissions will be compared to verify whether the US has achieved its NDC or not.

Similar comparably simple and concise examples can be given for other countries and/or Parties to the UNFCCC like China, India, or the European Union.

Our analysis models the emissions reductions projected under the totality of these NDCs, taking into account their macro-economic interactions across regions. In our original submission we already provided an overview of all interpretations made in relation to each NDC in the Supplementary Data. This extensive information remains also part of the revised submission.

The reviewer inquired about the precision we attribute to our results. The most important outcome from our analysis are the insights, not the numbers, and these insights we consider to be robust. Precision of a single scenario result is to be seen in the context of the uncertainty range we explore, a range of roughly 15 GtCO₂e/yr in 2030 or +20% to -10% around the median estimate. Again, here the distinction between projections and predictions has to be made. As highlighted in the response to the previous point, our research question is not to provide an as precise estimate of 2030 emissions as possible in the context of NDCs and national or sectorial policies.

We appreciate the reviewer's comment and its implications for increased clarity, and have made sure to clarify these aspects in the revised manuscript.

(3) The appropriateness of regional aggregation in our modelling approach.

Thank you for this question. This is an excellent comment.

Any NDC modelling exercise has to make trade-offs between the detail in the representation of national policies and the inclusion of global feedbacks and feedbacks between regions. Our research question and analysis explores global and regional uncertainties for which the representation of these feedbacks is crucial. Despite not having the granularity of a national model analysis, our approach allows us to explore global feedbacks much better than 'stacked' bottom-up assessments of national models. At the same time, our approach is also able to capture NDC assessments at the national level and their uncertainties to an appropriate degree (see more below).

Furthermore, the reviewer seemed confused by our technical explanation of how NDCs of single countries are aggregated into regional constraints for our modelling framework. The "some sort of calculations" the reviewer refers to turn out to be straight-forward and transparent arithmetic operations which ensure that NDCs of various countries, which can be defined in different ways, are aggregated correctly at a regional level.

To understand how our analysis performs for the assessment of individual countries, we carried out a dedicated set of experiments. In particular, we developed and implemented a scenario protocol in which, for one specific interpretation of the uncertainty dimensions, each of the roughly 140 NDCs was individually and incrementally added to its corresponding region in the MESSAGE model, and this in descending order of their emissions share in 2010. Comparing the incremental emission changes allows us to show how single-country NDCs influence regional emissions projections, and thus provides a first order estimate of the effect of single NDCs in our modelling framework, still, however, while taking into account macroeconomic linkages. The figure inserted below (Figure R1) illustrates that single NDCs can be represented and reflected, even when macro-economic dynamics are modelled at a more aggregate regional scale.

At the same time, the figure only shows the incremental decomposition of one of the 144 NDC interpretations which we modelled in our full analysis. The results shown in this figure should thus be seen in light of the significant uncertainties that we highlight in our analysis, amounting globally to about 15 GtCO₂e/yr. This is illustrated in Figure R2 which applies relative regional

uncertainties to the estimates of the 16 largest emitters and compares these to the ranges reported in the 2016 UNEP Emissions Gap Report and the INDC fact sheets of the University of Melbourne. This comparison shows that our national level model results and uncertainty ranges are very consistent with bottom up studies.

Figure R1 | Incremental reductions from a no-climate-policy baseline for each NDC per region. Each bar represents the total emissions per region after including one additional NDC. The difference between consecutive bars thus illustrates the influence of NDCs of individual countries.

Figure R2 | Incremental reductions from no-policy reference levels in 2030 in the IIASA IAM framework (blue features) compared to literature values from UNEP (2016) and the University of Melbourne (Meinshausen, 2015). The 'selected illustrative case' from this study assumes an SSP2 socioeconomic development, unconditional NDCs, PRIMAPHIST historical emission inventories, direct equivalence energy accounting, and does not count non-commercial biomass towards renewable energy. The variations found in the literature fall well within our uncertainty range. Furthermore, clearly different default assumptions are applied by the assessments of the different studies. Understanding these differences will be of important in future assessments of NDCs.

References

Meinshausen, M. (2015). *INDC Factsheets*. from Australian-German Climate and Energy College / University of Melbourne <http://climate-energy-college.net/indc-factsheets>
UNEP. (2016) *The Emissions Gap Report 2016*. (UNEP, Nairobi, Kenya, 2016).

Another concern with this paper is that the figures are too hard to read and make sense. For example, in Figure 1a, what are the individual lines? How many scenarios were analysed in the paper? What's the point being made in Figure 1b in the lines with dots? Figure 2 can easily be converted into a table or may be a simple bar-graph. Figure 3 is not clear at all.

RESPONSE: We thank the reviewer for his/her comments suggestions on these issues, and have made sure to clarify this in either the figures or their accompanying captions. The line with the dots/symbols in Figure 1b is a cumulative frequency distribution, the point of which is to illustrate the cumulative distribution and relative overlap of the three scenario subsets highlighted in this panel. It is a matter of taste whether one prefers cumulative distributions or histograms. We appreciate that the reviewer prefers the latter.

Our original manuscript already included the requested data of Figure 2 in table format, i.e. Table 2 in the main manuscript. This was mentioned explicitly in the accompanying table caption.

For further visual improvements of figures or text we have diligently followed the guidance received by the editor.

Why is there a discussion of carbon prices? If it is to show near-term versus long-term tradeoffs, millions of papers have shown that already. What is important for decision makers is how many nuclear power plants need to be built? What's the rate of change in the energy system transitions (see for example papers by van Vuuren et al. in the Focus Issue on cumulative emissions in *Environmental Research Letters*. There are also some interesting papers on near-term versus long-term tradeoffs in that Focus Issue). In what way is this "new science"? Also, in Figure 3b, what is shown is 2030 carbon price with respect to 2030 emissions. What can we conclude about near-term versus long-term from that figure?

RESPONSE: Carbon prices are a very common metric to compare the level of climate mitigation effort in the scientific climate policy literature (for example, see Chapter 6 in the most recent assessment report of the Intergovernmental Panel on Climate Change – IPCC, available at: <http://www.ipcc.ch/report/ar5/wg3/>). It is also a metric widely used by policymakers for planning purposes, for example, in the UK, and as illustrated by recent reports from the National Academy of Sciences where alternative costs of carbon are explored. We find providing a widely used point of comparison with the literature an important aspect of any robust scientific analysis, and are surprised that the reviewer does not share this view.

The common use of carbon prices as a mitigation metric was also appreciated by the two other expert reviewers, who did not consider this an odd choice and had no problems understanding the figures. To be sure, the figure shows emissions, carbon prices, and consumptions losses. Economic metrics for both the pre and post-2030 period are given, showing how different interpretations of the INDCs result in significant variations in the discontinuity in 2030 for limiting warming to 1.5°C or 2°C. While we are well aware of papers which discuss the trade-offs between near- and long-term action (and already cited four of them in our original manuscript), our current analysis looks at the implications of the range of emissions estimates derived from

the current INDCs. We can impossibly discuss the intertemporal trade-offs of the INDCs without providing these figures. Furthermore, we are also not aware of any publication that would already include such an illustration, and thus consider it a novel and important contribution to the body of literature.

At the same time, we agree with the reviewer that also different information could be provided. Given space constraints, it is unfortunately not possible to provide a full overview of energy system transitions as in the paper by van Vuuren et al in ERL. However, we now provide more context for the carbon prices that are shown by referencing the IPCC AR5, and also provide additional figures showing the upscaling of non-biomass renewables in the supplementary information (Supplementary Figure 3).

In terms of reorganizing figures and tables, I would recommend a major revision of the text and figures that starts describing the results to move from one figure to another to tell a well-knit story, which I think can be done.

RESPONSE: We appreciate the reviewer's suggestions on these editorial issues. At the same time, we also appreciate that another reviewer expressed that "the overall text is well written, and the logic is understandable", and yet a third indicated that the text was dense at times. Given these contradictory assessments, we will follow the *Nature Communications* editor's recommendations in this regard, as he/she has the best overview and expertise in this area.

A final concern I have is that that the authors do not in general seem to make their model available online. I tried my best to find it online and it just does not exist. How are we as scientific researchers to ensure reciprocity? It seems from this paper that the models MESSAGE and MAGICC are black boxes. Surprisingly, there is not even much documentation of the models in the Supplementary. Of course the final decision on this lies with the editor. Nevertheless, I thought I would flag it since this is an open source journal.

RESPONSE: Model documentation is already published elsewhere and is thus limited in the Supplementary Information. The reviewer seems to have missed this.

For instance, the MAGICC model is described in full detail in Meinshausen et al. *Atmospheric Chemistry and Physics* (2011), and can be accessed online on <http://live.magicc.org/>, where also download forms and links are available. The reference to the scientific and peer-reviewed model documentation was already provided in the original manuscript. This seems to be overlooked by the reviewer.

The MESSAGE model is described in Fricko et al. *Global Environmental Change* (2016), and its documentation can also be accessed online on <http://data.ene.iiasa.ac.at/message-globiom/> (a link updated from the one provided in our original manuscript). It seems that the reviewer also missed this.

The MESSAGE modelling framework is complex, but available upon request for research purposes to anyone interested in investing the necessary time. Examples of this are the use of MESSAGE by many research groups around the globe, including, for example, the University of Rio de Janeiro (e.g., Lucena et al. 2010, Nogueira et al. 2014), the Lithuanian Energy Institute (e.g., Norvaiša & Galinis 2016), the Malaysian Nuclear Agency (e.g., Fairuz et al. 2013) and University of Teheran (e.g., Shakouri & Aliakbarisani 2016). The wide-spread use of the MESSAGE modelling framework is in part owing to the fact that it is distributed by the

International Atomic Energy Agency (IAEA) to its member states as an energy planning tool and has been deployed in some 100 countries via this channel (<https://www.iaea.org/OurWork/ST/NE/Pess/capacitybuilding.html>).

References

- Lucena, A. F. P., Szklo, A. S., Schaeffer, R., Dutra, R. M. (2010) The vulnerability of wind power to climate change in Brazil. *Renew. Energy* 35, 904-912.
- Nogueira, L. P. P.; Lucena, A. F. P.; Rathmann, R.; Rochedo, P. R. R.; Szklo, A.; Schaeffer, R. (2014) Will thermal power plants with CCS play a role in Brazil's future electric power generation?. In *International Journal of Greenhouse Gas Control*, v. 24, p. 115-123.
- Norvaiša, E., Galinis, A. (2016) Future of Lithuanian energy system: Electricity import or local generation? *Energy Strategy Reviews*, 10, pp. 29-39.
- Fairuz SMC, Sulaiman MY, Lim CH, Mat S, Ali B, Saadatian O, Ruslan MH, Salleh E, Sopian K (2013) Long term strategy for electricity generation in Peninsular Malaysia - Analysis of cost and carbon footprint using MESSAGE. *Energy Policy* 62:493-502.
- Shakouri G H, Aliakbarisani S (2016) At what valuation of sustainability can we abandon fossil fuels? A comprehensive multistage decision support model for electricity planning. *Energy* 107:60-77.

SECOND REFEREE REPORT

Review Comments on NCOMMS-16-20105-T

We thank the reviewer for his/her constructive and clear comments. We have answered to each of them in detail below.

Understanding the origin of Paris Agreement emission uncertainties

This paper provides the systematic estimation of emission uncertainty of Paris agreement and its implications for the long-term climate policy. An integrated assessment model is a core of the methodology. Hundreds of scenarios associated with six uncertainty dimensions are produced. The authors show that 47-63 GtCO₂e is the range of 2030's global GHG emissions. This range has critical implications for the feasibility and costs to limit warming to well below 2 or 1.5 °C. The main source of uncertainty to determine 2030's global GHG emissions is socioeconomic assumption where they used Shared Socioeconomic Pathways (SSPs). They finally suggest how to decrease the uncertainty.

The overall text is well written, and the logic is understandable. However, there are several concerns before publishing. Here, I listed some points that could be modified or improved.

1) In the final section, the authors stress that the clarifying socioeconomic assumptions largely contributes to reducing uncertainty. That would be true from this model exercise. However, socioeconomic assumptions are abstract for policy implementation. Looking at the SSPs framework (Riahi, 2016), there are various elements that determine socioeconomic assumptions. Do authors ask to specify all of them in NDCs? I envisage that all countries cannot do such things realistically speaking. Then, I come up with another question whether the authors implicitly mentioned about some of the socioeconomic elements?

RESPONSE: This is a great comment. We do indeed vary the full set of elements that determine the socioeconomic assumptions (including GDP projections, population projections, technology development, etc.). How these assumptions were implemented for the SSPs in the MESSAGE framework is documented in Fricko et al (in press) *Global Environmental Change* (cited in the main manuscript of the paper).

The comment by the reviewer indicating that one can impossibly expect all SSP elements to be specified in NDCs is very valid, and we have followed up on this issue by more clearly indicating which NDC aspects could and which could not be easily improved. See also our response to the second comment below.

2) Following up to the above question, let us assume that socioeconomic assumptions are clarified in NDCs as authors suggest. Then, can we derive the same implications of the long-term mitigation goal addressed in this paper? For example, given the situation where accidentally all countries submit SSP2 conditions in NDCs, GDP, population, energy policies and so on until 2030 exactly follow SSP2 assumption. The uncertainty must be reduced significantly. However, don't we still have a possibility the world goes SSP3 direction which is worst socioeconomic assumptions regarding climate mitigation afterward, and the long-term implication is still uncertain? I may be able to agree that the socioeconomic condition is the main source of INDC emissions in 2030. However, I still feel that there is a gap from this 2030's emissions uncertainty and long-term implication.

RESPONSE: Thank you for this comment. This is a quite fundamental issue. After discussions with colleagues and stakeholders, we fully agree with the reviewer's remark that all countries can impossibly specify all aspects that drive emissions or other aspects in the SSPs. Calling for countries to specify these as a means to reduce uncertainty would thus likely not be very effective.

We have therefore rethought our recommendations to countries/policymakers. In our concluding section, we now describe that not all uncertainties are created equal. We identify two types of uncertainty. One type of uncertainty can be reduced through simple, technical clarifications (for example, which energy equivalence method is assumed, whether non-commercial biomass is considered a renewable energy carrier, or clear guidelines for NDC specifications). Other uncertainties, however, are not merely technical but the result of politically valid choices by countries (for example, defining an NDC as an intensity improvement target or as a relative improvement compared to a baseline). Countries can choose to continue to specify their NDC in such a way and given the uncertainty in socioeconomic development that influences the emissions outcomes of such NDCs, the uncertainties resulting from this choice will in that case be irreducible.

The reviewer will notice that we have adapted the concluding section of our paper to reflect this new framing.

3) The methodological description is not sufficient regarding attribution of uncertainty. I could guess that the authors run the all possible combinations of each uncertainty dimension something like $2 \times 3 \times 2 \dots$. However, there is no explanation how to attribute the uncertainty from such large number of scenarios. That should be documented elsewhere.

RESPONSE: Thank you for highlighting this. This was indeed not clearly expressed in our original submission. We now highlight our methodology for assessing the variation per uncertainty dimension in the Methods section and further in Supplementary Information (Supplementary Text 6).

In short, for a given uncertainty dimension we compare like-with-like scenarios (that is, scenarios which are identical in all but the uncertainty dimension of interest) and estimate the maximum variation for each of these small subsets.

4) The uncertainty regarding land use change emissions seems very important element to understand the uncertainty of future emissions. For example, IPCC AR5 WG3 fig 11.6 presents the range of current emissions is around 5 GtCO₂ and cannot be ignored. Then, how can we interpret the outcome of this study considering such additional uncertainty?

RESPONSE: Land-use uncertainty is an important factor indeed, which in some cases would be additional to the uncertainty we assess here. Unfortunately, we do not have the modelling capability at this moment to assess this uncertainty dimension in detail (which we fully acknowledge in our manuscript). Uncertainties in land-use emissions are not only due to emissions inventories, but also due to the accounting rules applied to land use, land-use change and forestry emissions and how they allow industrial emissions to vary. We fully appreciate the importance of the uncertainty surrounding the land-use contribution. We already included this in our original version but have now elaborated this a bit more in our revised version to highlight the potential important influence of this uncertainty.

5) How does the regional classification of the integrated assessment model influence to this study? For example, if you have single independent US, China, India, Russia, and Brazil? It would be nice to discuss.

RESPONSE: This is an important issue, also highlighted by another reviewer. We have carried out a dedicated sensitivity analysis to understand better how our model performs. Specifically, we developed a supplementary scenario protocol in which each of the roughly 140 NDCs is incrementally added to its respective region (and this assuming one consistent interpretation of the uncertainty dimensions). This allows us to understand how the inclusion of a single additional NDC into our framework affects our emissions estimates.

Three insights are particularly noteworthy. First, many regions are de facto dominated by one single country, be it China (CPA), the US (NAM), the EU (EEU + WEU), or Brazil (LAM). Also the reductions in those regions are thus dominated by the NDC of those countries. In the analysis we carried out for this paper, we start from the NDC definition of the largest emitter for each region (see Figure R1 below). Therefore, our NDC assessment represents the NDCs of the larger emitters as closely as possible, while it differs more for smaller countries within regions. Second, due to the weak NDCs of some countries, inclusion of additional NDCs sometimes results in an emissions increase within regions. This dynamic is the result of implicitly allowing trading of mitigation actions within regions. Three, extracting values for single countries, including their uncertainties (which we derived from the variations within regions) shows for the major emitters that our results nicely bracket the ranges of earlier assessments in the literature (see Figure R2 below).

Figure R1 | Incremental reductions from a no-climate-policy baseline for each NDC per region. Each bar represents the total emissions per region after including one additional NDC. The difference between consecutive bars thus illustrates the influence of NDCs of individual countries.

Figure R2 | Incremental changes from no-policy reference levels in 2030 in the IIASA IAM framework (blue features) compared to literature values from UNEP (2016) and the University of Melbourne (Meinshausen, 2015). The 'selected illustrative case' from this study assumes an SSP2 socioeconomic development, unconditional NDCs, PRIMAPHIST historical emission inventories, direct equivalence energy accounting, and does not count non-commercial biomass towards renewable energy. The variations found in the literature fall well within our uncertainty range. Furthermore, clearly different default assumptions are applied by the assessments of the different studies. Understanding these differences will be of important in future assessments of NDCs.

References

- Meinshausen, M. (2015). INDC Factsheets. from Australian-German Climate and Energy College / University of Melbourne <http://climate-energy-college.net/indc-factsheets>
- UNEP. (2016) The Emissions Gap Report 2016. (UNEP, Nairobi, Kenya, 2016).

THIRD REFEREE REPORT

Reviewer #3 (Remarks to the Author):

Thank you very much for the constructive suggestions. We have taken into account all suggestions and respond to each of them individually below.

Overall

The role of adapting to the effects of climate change are important, however, missing from most NDCs. Might be good to make clear that the focus will be rather on mitigation/emissions numbers. You hint at this in the limitations of the study later, but could be good to bring it up front somewhere.

RESPONSE: We fully agree and highlight this issue now upfront in our manuscript. In the introduction we now write: "These so-called nationally determined contributions (NDCs) cover aspects of mitigation and adaptation, together with issues related to means of implementation (for example, capacity building, international finance and technology transfer), comparability and fairness, or sometimes linkages to sustainable development.⁴ Here we focus on understanding their mitigation aspects."

With reference 4 referring to the UNFCCC Synthesis Report on the INDCs.

Not always easy to read at points – with dense information. The key messages or what was important from the study therefore doesn't come out so clearly at times. Although I caveat this comment because I don't know the journal nor its audience very well.

RESPONSE: We appreciate the reviewer's comment and agree that readability is key for a journal with quite a broad readership. In our revised version we have therefore attempted to make the text slightly less dense (shorter sentences) and guide the reader more in the general logic of the paper (see also points further below).

Unclear how some uncertainties were calculated which are one of the key parts to this paper. E.g. Page 6 lines 21. 23. This seems to 'jump ahead' a few steps in the logic from INDC assumption to translating into tCO_{2e} ranges. Would be good to know those steps, perhaps quickly in a flow/chart to explain your steps.

RESPONSE: This issue was also highlighted by another reviewer and thus clearly required some attention. In our revised manuscript we now include a schematic of the scenario protocol (see also below) and a more detailed explanation on of how uncertainty ranges are derived in Supplementary Text 6. We refer to this in the main text but unfortunately could not include much additional explanation in the main manuscript because of space constraints.

Figure R3 | Overview of scenario structure to explore six uncertainty dimensions listed in Table 1. A total of $3*3*2*2*2*2 = 144$ scenarios has been developed.

[REDACTION]

How do the pathways to 2020, 2030 fit with the overall IPCC expectations? Wasn't clear to me in the discussions: e.g. Page 8, line 20, where you explain that only below 44 Gt in 2030 will mean 1.5C is possible.

RESPONSE: Thank you very much for this comment. IPCC, in its Fifth Assessment Report, provides an assessment of pathways for limiting global CO₂-equivalent concentrations to around 450 ppm in 2100 (range 430-480 ppm). This category of scenarios is also assessed to be consistent with limiting global-mean temperature increase to below 2°C with greater than 66% probability. For the question of pathways that limit warming to 1.5°C, the IPCC provides only limited information, in particular, no annual benchmarks for 2020 or 2030 are reported. Here we can compare our values to the assessments of the UNEP Emissions Gap Reports, which provide this information.

We have included such a comparison in the revised version of the manuscript.

Are negative emission scenarios potentially needed for 1.5/2C. I only ask, not to highlight such technological needs (because it is clearly better to have a situation without them), but readers might expect a sentence or paragraph on it.

RESPONSE: The reliance on negative emissions technologies for the achievement of stringent mitigation scenarios is indeed what one could currently call a "trending" topic. We fully agree that there is great general interest in issues related to negative emissions. In our revised manuscript we therefore highlight how varying levels of 2030 emissions influence the reliance on such technologies and also that all our scenarios that limit warming to 2°C allow the use of such technologies.

How are emissions from peat fires accounted in the model, or other non-sovereign emissions from aviation/shipping? My work on this showed that it is large (e.g. >1 Gt p.a.) and difficult to measure. You hint at this in paragraph on p10 line 9.

RESPONSE: Emissions from peat fires are indeed large, and also a large source of uncertainty. Our model's strength is in its representation of the energy system and macro-economic effects between regions with respect to fuel and commodity prices. The treatment of land-use-related emissions, including peat fires or other sources, is not one of these strengths. We have no other way of easily taking into account these aspects, and thus clearly highlighting and acknowledging these limitations is the most transparent way forward. In our revised manuscript we have further elaborated the uncertainties related to land-use emissions and explicitly mention the uncertainty due to peat-land emissions with a reference to a recent overview publication on this topic (Harris et al, 2012, Science).

Specific points

Line 9. 'will be taking place' by whom? UNFCCC? Clarify

RESPONSE: By the Parties to the Paris Agreement, as part of the framework set out in the Agreement. This is slightly different from saying "under the UNFCCC" or "by the UNFCCC", as the Global Stocktake will be carried out by those countries that ratified the Paris Agreement (125 to date: http://unfccc.int/paris_agreement/items/9444.php), which can be fewer than the total amount of Parties to the UNFCCC (about 190). We have edited this for clarity in the text.

Line 26. 'and their conditions' ☒ Expand as is an important point. I see some in the supplement text but it could also be in the main text. Also dependent on which models and assumptions are applied. It's all pretty subjective which might highlight the differences. Indeed, some analysis takes INDCs at 'face value' rather than making own assumptions on GDP growth of a country (specifically, India). I don't see a discussion of GDP assumptions in the text nor supplement for instance. In Table 1, It would be good to know how GDP assumptions are accounted.

RESPONSE: Thank you for raising this comment. The "and their conditions" text here refers to the studies underlying the range found in an earlier study. While we cannot discuss the GDP assumptions of these underlying studies in detail, we can do so for our own study. We include a direct reference to the study underlying our socioeconomic assumptions in Table 1.

Accompanying this reference, all underlying GDP assumptions have been made available in full (and at a country level) online (<https://tntcat.iiasa.ac.at/SspDb/>). In our analysis, we do not take NDCs at face value, but explore how varying GDP projections influence the potential emissions outcomes.

Page 6. Line 21. How is 10-20% uncertainty on international finance provision calculated?
Likewise line 23.

RESPONSE: This sentence was not formulated very clearly and obviously had the potential to confuse the reader. The 10-20% uncertainty refers to the change in global emissions outcome under the assumption that all provisions for moving to the conditional end of the NDCs are met. It is calculated by pair-wise comparing the emissions levels that are projected under full implementation of all unconditional NDCs with projections of global emissions for the conditional case, all other aspects remaining the same.

Reviewers' comments:

Reviewer #1 (Remarks to the Author):

I appreciate the response to my rather critical comments to the manuscript by Rogelj et al. I think the authors have clearly defended the novelty of the research question and the associated methodology. The authors have clearly undertaken a good deal of effort to respond to my criticisms regarding regional aggregation in the model they have used in their analysis. Well done and thank you. The paper is now several notches above where it was originally.

However, to strengthen your paper further and to make it more accessible to the broader community, I suggest the following additional revisions before publication.

1. Include a summary of your discussion in response to the point about the research question this paper is trying to answer, and its novelty – this can be summarized based on your response in the response to reviewers document. This should go into the supplementary material. Do highlight the points about why you did not model actual policies and why it does not matter or is not relevant to this exercise. Highlight that modeling of actual policies to achieve the NDCs will be a part of future modeling efforts to address a different set of questions – related to this study but possibly different.

2. Include Figure R2 in Supplementary along with the discussion on why alternative regional disaggregations would not affect the qualitative insights of the study. I still think that this is a major limitation of the study – that you use a model with lesser regional detail when models with more detail are available (some of which, I believe are even open source). You should definitely comment on how your results might or might not change if such other models are employed. This is not to say that models with more details are better, but your choice of the modeling tool for the research question you ask certainly begs the question of why a model with more detail was not used or results compared against, especially when such tools are available in the public domain? This discussion should go into the supplementary in order to avoid overconfidence in your results.

3. Be consistent throughout the main text and supplementary material on the use of "NDC" versus "INDC".

4. Clarify, via a footnote or additional text in the caption of Figure 3b that the dashed and dotted-dashed lines show post-2030 carbon prices even though the y-axis is labeled "2030 carbon price". One quick idea is to just remove "2030" from the axis label and call out what the thick and dashed lines represent in the figure (in addition to explaining in caption). I think the y-axis in Figure 3c does not say 2030. Please be consistent across the b and c panels in whatever step you take to clean this figure.

Thank you also, for pointing to <http://live.magicc.org/> - I am aware of this but the source code is not available to the best of my knowledge. I would still alert this to the editor and leave it to the editor's discretion.

Thank you for the clarification about the MESSAGE model.

Well done overall and looking forward to the paper being published.

Reviewer #2 (Remarks to the Author):

I greatly appreciate the response of the authors. I mostly satisfy with them. However, two things regarding land use are raised. First, I oppose to the authors' response that

they do not have the modeling capability because I believe that there is one of the most advanced land use models in the world called GLOBIOM in the IIASA IAM modeling framework. Second, the authors added the description that "peat drainage and burning are often excluded from the estimates" in this revision, and it seems one of the reasons why they don't take into account. However, at least Indonesian INDC considers the peat related emissions and their reduction contribution in their ways, and that is not a small emission source. I can understand that there are accounting issues and challenges in the modeling (even for GLOBIOM), but if so, I wonder authors could show the range of the uncertainty due to the land use related emissions even with a rough method. Or if authors can insist that land-use change emissions are not significant and the conclusion would not be affected by them, it would be fine.

We thank the referees for accepting our earlier revisions, and for highlighting the additional points which we address in this revision. We have considered all referee comments.

Point-by-point responses to the referee comments are inserted below in blue.

INDIVIDUAL RESPONSES TO THE REFEREES

FIRST REFEREE REPORT

Reviewer #1 (Remarks to the Author):

I appreciate the response to my rather critical comments to the manuscript by Rogelj et al. I think the authors have clearly defended the novelty of the research question and the associated methodology. The authors have clearly undertaken a good deal of effort to respond to my criticisms regarding regional aggregation in the model they have used in their analysis. Well done and thank you. The paper is now several notches above where it was originally.

However, to strengthen your paper further and to make it more accessible to the broader community, I suggest the following additional revisions before publication.

RESPONSE: Thank you for the positive assessment of our revised manuscript. We are glad that our efforts were able to convincingly respond to the criticisms raised. Below we respond to each of the remaining points.

1. Include a summary of your discussion in response to the point about the research question this paper is trying to answer, and its novelty – this can be summarized based on your response in the response to reviewers document. This should go into the supplementary material. Do highlight the points about why you did not model actual policies and why it does not matter or is not relevant to this exercise. Highlight that modeling of actual policies to achieve the NDCs will be a part of future modeling efforts to address a different set of questions – related to this study but possibly different.

RESPONSE: Thank you for this suggestion. A summary of the text describing our research question and the modelling framework strengths and limitations was included in the supplementary material as Supplementary Note 1.

2. Include Figure R2 in Supplementary along with the discussion on why alternative regional disaggregations would not affect the qualitative insights of the study. I still think that this is a major limitation of the study – that you use a model with lesser regional detail when models with more detail are available (some of which, I believe are even open source). You should definitely comment on how your results might or might not change if such other models are employed. This is not to say that models with more details are better, but your choice of the modeling tool for the research question you ask certainly begs the question of why a model with more detail was not used or results compared against, especially when such tools are available in the public domain? This discussion should go into the supplementary in order to avoid overconfidence in your results.

RESPONSE: Thank you for this suggestion. We agree that such a discussion can be very valuable to the reader, and have included both figure and discussion in Supplementary Note 5.

3. Be consistent throughout the main text and supplementary material on the use of "NDC" versus "INDC".

RESPONSE: Thank you. We changed all instances to "NDC", adding the intended "I" only when explicitly applicable.

4. Clarify, via a footnote or additional text in the caption of Figure 3b that the dashed and dotted-dashed lines show post-2030 carbon prices even though the y-axis is labeled "2030 carbon price". One quick idea is to just remove "2030" from the axis label and call out what the thick and dashed lines represent in the figure (in addition to explaining in caption). I think the y-axis in Figure 3c does not say 2030. Please be consistent across the b and c panels in whatever step you take to clean this figure.

RESPONSE: Thank you for highlighting this potential source of confusion. As suggested, we have removed the "2030" from the label while explaining the lines in both legend and caption. We applied the same changes to Supplementary Figures 1 and 2, for consistency.

Thank you also, for pointing to <http://live.magicc.org/> - I am aware of this but the source code is not available to the best of my knowledge. I would still alert this to the editor and leave it to the editor's discretion.

Thank you for the clarification about the MESSAGE model.

RESPONSE: We are glad that our clarifications were useful.

Well done overall and looking forward to the paper being published.

RESPONSE: Thank you.

SECOND REFEREE REPORT

Review Comments on NCOMMS-16-20105-T

I greatly appreciate the response of the authors. I mostly satisfy with them.

RESPONSE: Thank you for the positive assessment of our revised manuscript. We respond to the last remaining issue below.

However, two things regarding land use are raised. First, I oppose to the authors' response that they do not have the modeling capability because I believe that there is one of the most advanced land use models in the world called GLOBIOM in the IIASA IAM modeling framework. Second, the authors added the description that "peat drainage and burning are often excluded from the estimates" in this revision, and it seems one of the reasons why they don't take into account. However, at least Indonesian INDC considers the peat related emissions and their reduction contribution in their ways, and that is not a small emission source. I can understand that there are accounting issues and challenges in the modeling (even for GLOBIOM), but if so, I wonder authors could show the range of the uncertainty due to the land use related emissions even with a rough method. Or if authors can insist that land-use change emissions are not significant and the conclusion would not be affected by them, it would be fine.

RESPONSE: We fully agree with the reviewer's concerns related to land use and NDCs. This is an important additional dimension and we have made further efforts to clarify its potential impact on our results. A spatially explicit NDC assessment in the detailed GLOBIOM framework is not possible at this point. Therefore, we took another path and brought together literature estimates of historical regional land-use emissions (FAOSTAT 2016), land-use NDC estimates and their uncertainty range (Forsell et al. 2016), and our own uncertainty estimates per region. The result of this comparison is now shown in the figure below (also included in the supplementary information to the paper). We show that in some regions land-use emissions represent a large share of the total emissions, and also put the assessed uncertainties in LULUCF contributions to NDCs in context of the other uncertainties at the global and regional level. Forsell et al. (2016) provide detailed data and uncertainties for selected countries, which we map onto 5 regions of our framework, as an illustration of the potential effect.

Supplementary Figure 4 | Illustration of potential influence of land-use emissions on NDC uncertainties. **a**, Share of year-2010 land-use emissions and removals as percentage of total regional emissions. Regional definitions are given in Supplementary Table 2. Land-use emissions include both emissions and removals as reported in FAOSTAT (2016) (fields: “land use total” and “Net emissions/removals (CO₂eq)”). They are compared to the total regional GHG emissions in the MESSAGE model; **b**, estimates of the magnitude of uncertainty induced in 2030 per source relative to the median estimate, with the uncertainty in land use, land-use change, and forestry (LULUCF) contributions taken from Forsell et al (2016) and indicated by the blue circle. The blue circles show the relative magnitude of the emissions uncertainty range for single countries reported in Table 3 of Forsell et al. Forsell et al noted that many NDCs do not contain specific targets for the LULUCF contributions. The estimates shown here thus only give a first comparison: they do not represent a full assessment of LULUCF uncertainty and they also cover only a limited set of countries. Finally, uncertainty in the LULUCF part of NDCs does not have to translate in uncertainty of the full NDC. For example, in the case of the US, the LULUCF contribution of its NDC comes with important uncertainties. However, the overall economy-wide target of its NDC is not affected by this as it applies to all sectors and is relative to a historical base year. Under the US NDC, a shortfall in mitigation in the LULUCF sector should thus be balanced by deeper reductions in other sectors.

References:

Forsell, N., O. Turkovska, M. Gusti, M. Obersteiner, M. d. Elzen and P. Havlik (2016). "Assessing the INDCs' land use, land use change, and forest emission projections." *Carbon Balance and Management* 11(1): 26.

FAOSTAT. FAOSTAT Emissions Database. February 8, 2016 ed. Rome: Food and Agricultural Organization of the United Nations; 2016. <http://faostat3.fao.org/home/E>

REVIEWERS' COMMENTS:

Reviewer #2 (Remarks to the Author):

No, I have no oppositions to the publication of this paper.
I appreciate the authors great work.